# Expansion and contraction of resource allocation in sensory bottlenecks

**Laura R Edmondson[1,2], Alejandro Jiménez Rodríguez[2,3], Hannes P Saal[1,2]\***

[1]Active Touch Laboratory, Department of Psychology, University of Sheffield, Sheffield, United Kingdom; [2]Sheffield Robotics, University of Sheffield, Sheffield, United Kingdom; [3]Department of Computer Science, University of Sheffield, Sheffield, United Kingdom

**Abstract** Topographic sensory representations often do not scale proportionally to the size of their input regions, with some expanded and others contracted. In vision, the foveal representation is magnified cortically, as are the fingertips in touch. What principles drive this allocation, and how should receptor density, for example, the high innervation of the fovea or the fingertips, and stimulus statistics, for example, the higher contact frequencies on the fingertips, contribute? Building on work in efficient coding, we address this problem using linear models that optimally decorrelate the sensory signals. We introduce a sensory bottleneck to impose constraints on resource allocation and derive the optimal neural allocation. We find that bottleneck width is a crucial factor in resource allocation, inducing either expansion or contraction. Both receptor density and stimulus statistics affect allocation and jointly determine convergence for wider bottlenecks. Furthermore, we show a close match between the predicted and empirical cortical allocations in a well-studied model system, the star-nosed mole. Overall, our results suggest that the strength of cortical magnification depends on resource limits.

## Editor's evaluation

The article develops a mathematical approach to study the allocation of cortical area to sensory representations in the presence of resource constraints. The theory is applied to study sensory representations in the somatosensory system. This problem is largely unexplored, the results are novel, and can be of interest to experimental and theoretical neuroscientists.

**\*For correspondence:**
h.saal@sheffield.ac.uk

**Competing interest:** The authors declare that no competing interests exist.

## Introduction

In many sensory systems, receptors are arranged spatially on a sensory sheet. The distribution of receptors is typically not uniform, but instead, densities can vary considerably. For example, in vision, cones are an order of magnitude more dense in the fovea than in the periphery (*Goodchild et al., 1996*; *Wells-Gray et al., 2016*). In the somatosensory system, mechanoreceptors are denser in the fingertips than the rest of the hand (*Johansson and Vallbo, 1979*). Alongside the density of receptors, the statistics of the input stimuli can also vary. For example, the fingertips are much more likely to make contact with objects than the palm (*Gonzalez et al., 2014*). Subsequent sensory areas are typically arranged topographically, such that neighbouring neurons map to nearby sensory input regions, for example, retinotopy in vision and somatotopy in touch. However, the size of individual cortical regions is often not proportional to the true physical size of the respective sensory input regions and, instead, representations might expand (often called magnification) or contract. For example, both the fovea and the fingertips exhibit expanded representations in early visual and somatosensory cortices, respectively, compared to their physical size (*Azzopardi and Cowey, 1993*; *Engel et al., 1997*; *Sereno*

*et al., 1995*; *Martuzzi et al., 2014*). What determines this cortical magnification? For somatotopy, it has been proposed that cortical topography might directly reflect the density of sensory receptors (*Catani, 2017*). On the other hand, receptor density alone is a poor predictor of magnification (*Corniani and Saal, 2020*) and work on plasticity has established that cortical regions can expand and contract dynamically depending on their usage, suggesting that expansion and contraction might be driven by the statistics of the sensory stimuli themselves (*Coq and Xerri, 1998*; *Merzenich and Jenkins, 1993*; *Xerri et al., 1996*).

Here, we tackle this problem from a normative viewpoint, employing efficient coding theory, which has been widely used to model and predict sensory processing. Efficient coding theory includes a number of different approaches such as sparse coding, redundancy reduction and predictive coding, depending on the constraints of the proposed problem (*Chalk et al., 2018*). Here, we focus on efficient coding via redundancy reduction (*Barlow, 1961*), which suggests that neural populations are tuned to maximize the information present in the sensory input signals by removing redundant information (*Atick, 1992*; *Atick and Redlich, 1990*; *Attneave, 1954*; *Graham and Field, 2009*; *Chechik et al., 2006*). Efficient coding models have been most prominent in vision (*Kersten, 1987*; *Karklin and Simoncelli, 2011*; *Atick, 1992*; *Atick and Redlich, 1990*; *Doi et al., 2012*; *Olshausen and Field, 1996*; *Olshausen and Field, 1997*; *Olshausen and Field, 2004*; *Bell and Sejnowski, 1997*) and audition (*Smith and Lewicki, 2006*; *Lewicki, 2002*). This prior work has mostly focused on predicting the response properties and receptive field structure of individual neurons. In contrast, here we ask how receptive fields—independent of their precise structure—should tile the sensory sheet when the receptors themselves differ in density and activation levels.

Some aspects of magnification in topographic representations have been qualitatively reproduced using self-organizing maps (*Ritter et al., 1992*). However, these models generally lack a clear cost function and the magnification factor can be determined exactly only in rare cases, while a general expression is lacking (*Ritter and Schulten, 1986*). Assuming a uniform density of output neurons, cortical maps may be optimizing for the spread of incoming information to be equally distributed over these neurons (*Plumbley, 1999*).

In contrast to receptor density, there has been some work on how populations of neurons should encode non-uniform stimulus statistics using Fisher information (*Ganguli and Simoncelli, 2010*; *Ganguli and Simoncelli, 2014*; *Ganguli and Simoncelli, 2016*; *Yerxa et al., 2020*). This approach aims to approximate mutual information and is used to calculate optimal encoding in a neural population (*Yarrow et al., 2012*) however, its use is restricted to specific conditions and assumptions (*Berens et al., 2011*; *Bethge et al., 2002*). Rather than receptive fields uniformly tiling the input space, the optimal population should be heterogeneous, with receptive fields placed more densely over high-probability inputs, at detriment to low-probability regions (*Ganguli and Simoncelli, 2014*). Our approach differs from this prior work: rather than maximizing information between the neural population and the stimulus itself, we instead consider information between the neural population and an initial population of receptor neurons. This places a limit on the total amount of information that can be represented.

The need for information maximization is often motivated by resource constraints. These can take the form of an explicit bottleneck, where the number of receptor neurons is greater than the number of output neurons. This is the case in the early visual system, where photoreceptors in the retina are much more numerous than the retinal ganglion cells to which they project (*Wells-Gray et al., 2016*; *Goodchild et al., 1996*). Other sensory systems might lack such explicit bottlenecks, but still place limits on the amount of information that is represented at a higher-order processing stage.

How then should resource allocation change for different-sized bottlenecks, given varying densities of receptors and different stimulus statistics? Here, we derive optimal neural allocations for different bottlenecks, while systematically varying receptor density and stimulus statistics. A preliminary version of these results restricted to the effects of receptor density in a 1D space was previously presented as a conference paper (*Edmondson et al., 2019*).

## Results

We restrict ourselves to linear models and only consider second-order statistics of the sensory signals, such that redundancy reduction simplifies to decorrelation (for examples from the visual literature, see *Hancock et al., 1992*; *Simoncelli and Olshausen, 2001*; *Doi and Lewicki, 2005*). We also introduce a

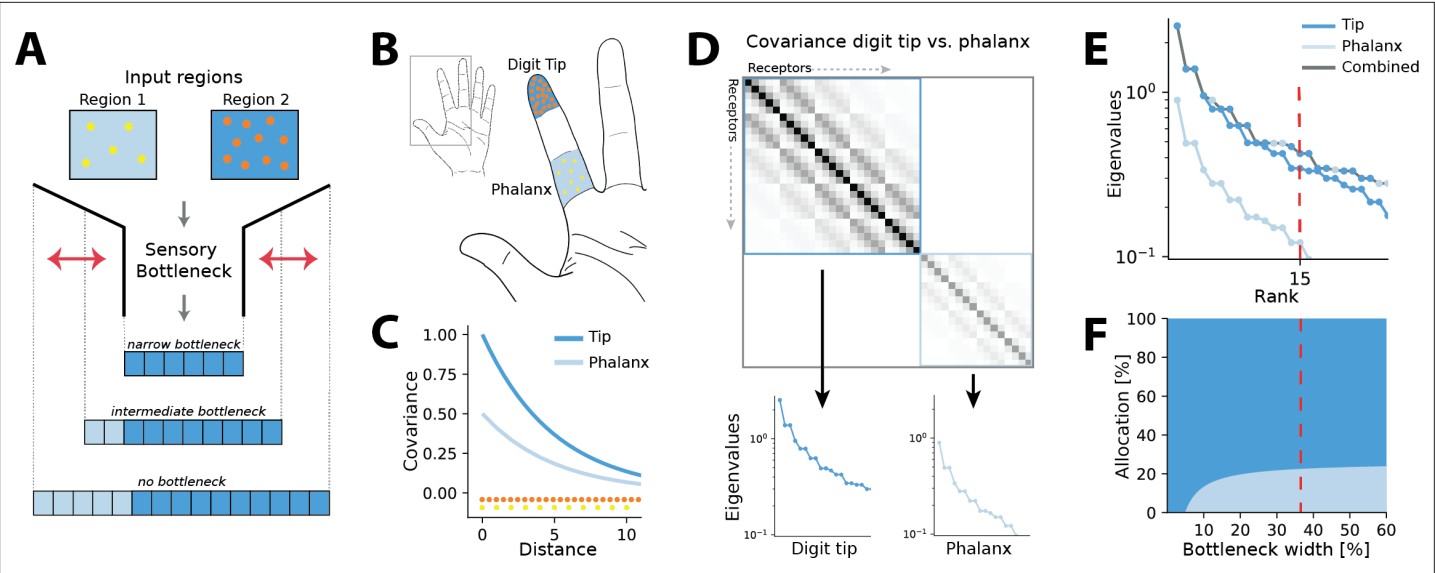

**Figure 1.** Illustration of the resource allocation problem and solution outline. (**A**) Abstract problem setup. When two regions vary in their receptor densities and activation, how should a shared resource be allocated between them when that resource is limited, as in a sensory 'bottleneck'? The allocation between the two regions may vary depending on the width of the bottleneck, for example, for narrow bottlenecks it might be most efficient to allocate all resources to the higher density region 2. (**B**) Application of the problem to the tactile system. The density of touch receptors differs across regions of the hand (e.g. fingertip, shown in orange, versus finger phalanx, yellow). Different finger regions make contact with objects at different rates (dark blue versus light blue shading, darker colours indicating higher contact rates). (**C**) Sensory inputs are correlated according to a covariance function that decays with distance between receptors on the sensory sheet. This function is evaluated at different receptor distances depending on the density of sensory receptors (orange versus yellow dots at the bottom). Regions with higher probability of activation exhibit greater variance (dark versus light blue curves). (**D**) Decorrelation of sensory inputs in the presence of a bottleneck is achieved by calculating and retaining the eigenvectors and eigenvalues of the receptor covariance matrix. Here, this matrix is approximated as a block matrix, which allows calculating the eigenvalues for each region individually (dark blue versus light blue box). (**E**) The combined set of eigenvalues from all regions is then sorted; the region where each successive eigenvalue in the combined sorted set originates from determines where that output neuron's receptive field will fall. (**F**) Counting which input regions successive eigenvalues belong to results in the allocation breakdown for different bottlenecks. For certain forms of the covariance function, this allocation can be calculated analytically.

sensory bottleneck, such that the number of output neurons is smaller than the number of receptors. Specifically, $r = Wx$, where $r$ is the $m$-dimensional output vector, $x$ is the $n$-dimensional input vector (with $n > m$) containing sensory receptor responses, and $W$ contains the receptive field. Assuming a low-noise regime, optimal decorrelation can be achieved if $W$ takes the following form:

$$W = \Lambda^{-\frac{1}{2}} \Phi^T, \tag{1}$$

where $\Lambda$ and $\Phi$ contain the (top $m$) eigenvalues and eigenvectors, respectively, of the covariance matrix $\Sigma = X^T X$.

In the specific setup we consider here, receptors are placed in non-uniform densities across the sensory sheet, with some regions having a higher density than others. For example, in *Figure 1B*, the digit tip is tiled more densely than the rest of the finger. The covariance of receptor activation decreases with distance (see *Figure 1C*). Thus, directly neighbouring units in denser regions covary more strongly than those from less dense regions. The activation level of receptors can also vary between regions, which is modelled through scaling the covariance function (see Appendix 1 for further details). We initially focus on negative exponential covariance functions, for which the eigenvalues and the resulting allocation can be calculated analytically. The covariance between receptors $x_i$ and $x_j$ for two regions $R_1$ and $R_2$, differing in density and activation, can thus be expressed as

$$\Sigma_{ij}^{(R1)} = e^{-\gamma|x_i - x_j|} \quad \text{and} \quad \Sigma_{ij}^{(R2)} = a e^{-d\gamma|x_i - x_j|}, \tag{2}$$

where $a$ denotes the receptor activation ratio and $d$ the receptor density ratio between both regions. It can be seen that changing the density affects the decay of the covariance function and thereby 'stretches' the space, while changing the activation scales the function.

We approximate the regions as separate, such that no receptors are activated from both regions simultaneously. The covariance matrix across all receptors then forms a block matrix, where the covariance between separate regions is zero (**Figure 1D**):

$$\boldsymbol{\Sigma} = \begin{bmatrix} \boldsymbol{\Sigma}^{(R1)} & \mathbf{0} \\ \mathbf{0} & \boldsymbol{\Sigma}^{(R2)} \end{bmatrix} \tag{3}$$

Thus, eigenvalues can be calculated separately from the covariance matrices of each region (inset panels in **Figure 1D**), and for negative exponential covariance functions, this can be done analytically (see 'Methods' for mathematical derivations):

$$R_1 : \lambda_{l,m}^{(R1)} = \frac{2\gamma}{\pi^2 L^{-2}(l^2 + m^2) + \gamma^2} \quad \text{and} \quad R_2 : \lambda_{n,o}^{(R2)} = \frac{2\gamma a \sqrt{d}}{\pi^2 L^{-2}(n^2 + o^2) + \gamma^2} \tag{4}$$

Finally, the resulting eigenvalues are ordered by magnitude for both regions combined (grey line in **Figure 1E**). A bottleneck is introduced by restricting the total number of output neurons. This is done by selecting from the combined set of ordered eigenvalues until the limit is reached. The proportion of eigenvalues originating from each input region determines its allocation for the chosen bottleneck width (see red dashed line in **Figure 1E and F**).

In the following, we will present results for the common case of 2D sensory sheets, while results for 1D receptor arrangements are summarized in Appendix 6. For ease of analysis, all cases discussed will assume two sensory input regions of equal size, which are differing in receptor density, input statistics, or both. In all our examples, receptors in the low-density (baseline) region are always spaced on a grid with a distance of 1.

## Resource limits determine the amount of magnification

First, we investigated resource allocation in bottlenecks for heterogeneous density of receptors and heterogeneous stimulus statistics separately, whilst keeping the other factor uniform across the input regions.

### Heterogeneous density

For two regions with different receptor densities, we found that the higher density region could either expand or contract relative to its input density, depending on the width of the bottleneck. Specifically, for narrow bottlenecks (smaller than approximately 10% of input neurons), the higher density region is either exclusively represented or its representation is expanded compared to a proportional density allocation (see example in **Figure 2A**). Mathematically, this can be explained by a multiplicative scaling of the eigenvalue function for the higher density region (see illustration in **Figure 2—figure supplement 3A**). In contrast, for intermediate bottlenecks, low-density regions expand their representation, beyond what is expected for a proportional mapping (see dashed red line in **Figure 2A** denoting proportional allocation), leading to a contraction of the high-density region. For negative exponential covariance functions, this allocation converges to a fixed ratio at wider bottlenecks of $1/(1 + \sqrt{d})$, where $d$ is the density ratio between both regions (dashed yellow line in **Figure 2A**; see 'Methods' for derivation), as neurons are allocated to either region at a fixed ratio (see inset in **Figure 2—figure supplement 3A**). Finally, for wide bottlenecks, all information arising from the low-density region has now been captured, and any additional output neurons will therefore be allocated to the high-density region only. The over-representation of the low-density region thus decays back to the original density ratio. The overall nonlinear effect of bottleneck width is present regardless of the ratio between the densities (see **Figure 2B**). The spatial extent of the correlations over the sensory sheet (controlled by the decay constant γ in the covariance function, see 'Methods') determines allocation at narrow bottlenecks, and how fast the allocation converges, but does not affect the convergence limit itself (**Figure 2C**). As γ increases, and therefore spatial correlations decrease, the convergence limit is approached only at increasingly wider bottlenecks. The extent of magnification thus heavily

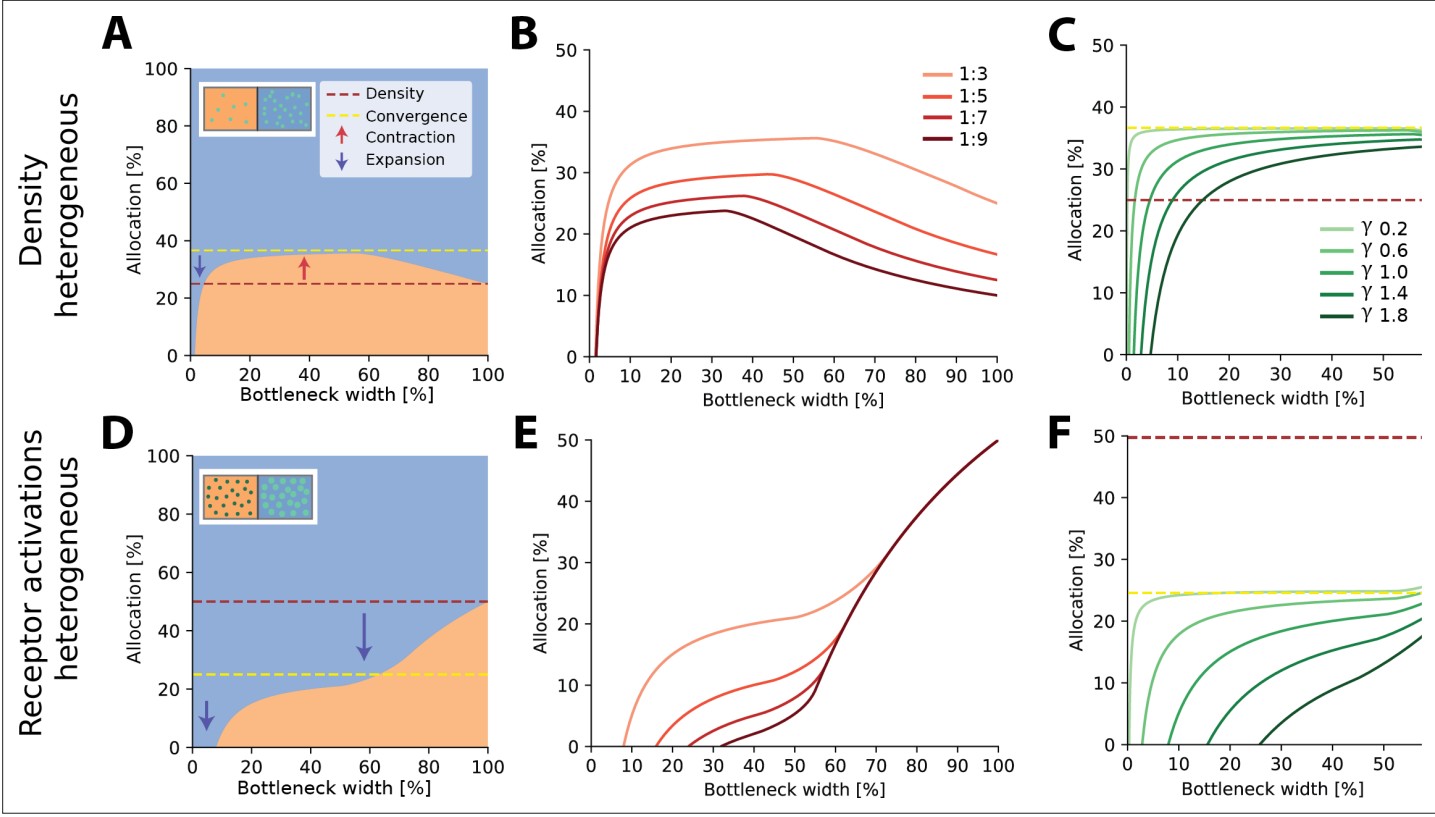

**Figure 2.** Optimal resource allocation for heterogeneous receptor densities or input statistics leads to complex trade-offs. (**A**) Illustration of resource allocation for heterogeneous receptor density but homogeneous stimulus statistics over all bottleneck sizes. Orange denotes the lower density region and blue the higher density region, with a ratio of 1:3. Dotted lines show proportional representation according to receptor numbers (red) and convergence of the optimal allocation in the limit (yellow). Arrows indicate contraction (up) and expansion (down) of the higher density region representation. Inset demonstrates change in density between the regions. (**B**) Bottleneck allocation boundaries for different density ratios (given as low:high). The area below each line corresponds to the low-density representation, while the area above corresponds to the high-density representation, as in (**A**). (**C**) Effect of changing the extent of the spatial correlations (parameterized by the decay value γ, see 'Methods' for details and *Figure 2—figure supplement 1* for an illustration of the covariance function for different values of γ). Density ratio is set at 1:3 for all γ. Increasing γ leads to expansion of the higher density region for a larger initial portion of the bottleneck. (**D–F**) Same as in row above but for homogeneous density and heterogeneous receptor activation ratios. (**D**) Illustrative example with the blue region having higher receptor activation. Note that the representation of the higher activation region is expanded for all bottleneck widths. Inset demonstrates difference in activation between the regions. Larger, brighter coloured points indicate higher activation for that region compared to the other. (**E**) Allocation boundaries for different activation ratios. The representation of the higher activation regions is expanded for all bottlenecks. As activation ratio increases, the highly active region allocation is expanded for wider bottlenecks. (**F**) Changing the extent of spatial correlations (γ) has larger effects when the activation ratio is heterogeneous (set at 1:3 for all γ) compared to heterogeneous density (**C**). See *Figure 2—figure supplement 2* for an equivalent figure considering 1D receptor arrangements.

The online version of this article includes the following figure supplement(s) for figure 2:

**Figure supplement 1.** Effect of different values of γ on the covariance function decay.

**Figure supplement 2.** Resource allocation for heterogeneous receptor densities and variations in input statistics in 1D.

**Figure supplement 3.** Illustration of eigenvalue sorting and resulting allocation.

**Figure supplement 4.** Limit on information rather than number of neurons.

depends on the correlational structure of the stimuli for narrower bottlenecks, while receptor densities are more important for wider bottlenecks.

## Heterogeneous statistics
Aside from receptor densities, the stimulus statistics can also vary over the input space, leading to differences in receptor activation levels between the regions and their associated response variance. Overall, allocations for heterogeneous receptor activation are similar to those found with

heterogeneous density. However, while the allocations are again a nonlinear function of bottleneck width, the representations are solely expanded for the higher activation region for all bottleneck widths (see example in *Figure 2D*). The extent of this expansion depends on the width of the bottleneck and is again more extreme for narrower bottlenecks (see *Figure 2E*). The convergence limit is $1/(1 + a)$, where $a$ is the activation ratio, implying that the level of expansion and contraction in intermediate bottlenecks is more extreme than in the heterogeneous density case (see difference between red and yellow dashed lines in *Figure 2A and D*). Finally, the effect of spatial correlations is also more pronounced (*Figure 2F*).

In the cases described above, the bottleneck was constrained by the number of output neurons. Alternatively, the limiting factor might be set as the amount of information (total variance) captured. Doing so results in allocation curves that retain the nonlinear behaviour described here (see *Figure 2—figure supplement 4* for illustrations). An additional consideration is the robustness of the results regarding small perturbations of the calculated eigenvalues. As allocation depends on the ranks of the eigenvalues only, low levels of noise are unlikely to affect the outcome for narrow bottlenecks, especially since eigenvalues are decaying rather steeply in this regime. On the other hand, allocation in wider bottlenecks is determined from small tail eigenvalues which are much more sensitive to noise (which is also evident when comparing the analytical solution to numerical ones). Allocation can therefore be expected to be somewhat less robust in those regimes.

In summary, we find that representations of different input regions can contract or expand, depending on the bottleneck width. This effect plays out similarly for differences in receptor density and receptor activation, however, with some crucial differences. Finally, for narrow bottlenecks, the spatial extent of the correlations across the sensory sheet becomes an important driver.

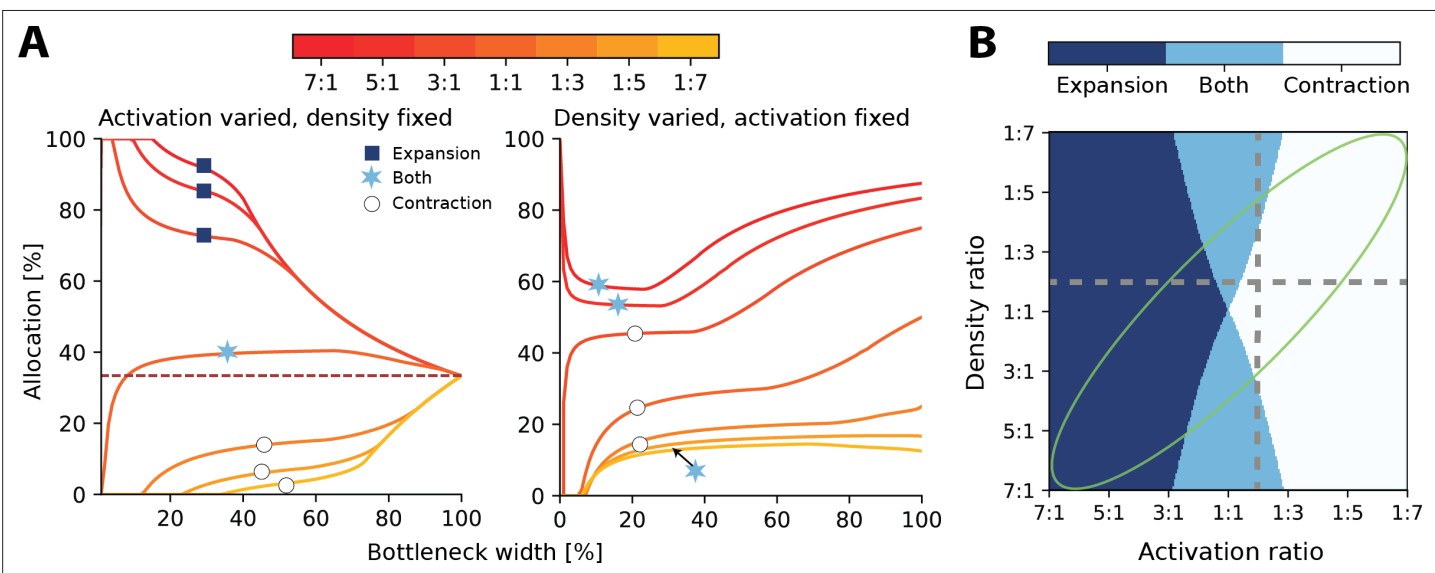

**Figure 3.** Interactions between heterogeneous statistics and density. (**A**) Allocations with both heterogeneous density and activation ratios. Expansion and contraction for a baseline region where relative density and activation is varied over the other region. All ratios given are baseline:other region. Left: fixed density ratio of 1:2, while activation ratio is varied between 7:1 and 1:7 (see colourbar). Purple dashed line indicates allocation proportional to density ratio. Right: fixed activation ratio of 1:2, while density ratio is varied between 7:1 and 1:7. The coloured markers indicate whether the baseline region is expanded (dark blue squares), contracted (white circles), or both (light blue stars) across all bottlenecks. (**B**) Possible expansion/contraction regimes for the baseline region based on combinations of density and activation ratios. Colours as shown by the markers in (**A**). Grey dashed lines show all possible allocation regimes for an example with either fixed density ratio (horizontal, corresponding plot in **A**, left) or activation ratio (vertical, **A**, right). When activation is fixed and density is varied, the allocations can be either expanded/both or contracted/both across the full bottleneck width. In contrast, when the density ratio is fixed and activation is varied, the allocation of a region could be any of the three regimes. The green ellipse highlights parameter combinations where activation and density ratios are correlated. See *Figure 3—figure supplement 1* for a comparison of how receptor density and activation interact between 1D and 2D receptor arrangements.

The online version of this article includes the following figure supplement(s) for figure 3:

**Figure supplement 1.** Comparison between 1D and 2D results for heterogeneous activation and density.

## Interplay between stimulus statistics and receptor density

In sensory systems, such as in touch, both the density and input statistics vary across regions and will therefore jointly determine the resulting allocation. As a consequence, the convergence for intermediate bottlenecks will depend on both density and activation ratios, and can be calculated as $\frac{1}{(1+a\sqrt{d})}$, where $a$ is the activation and $d$ is the density ratio (see 'Methods' for derivation).

At narrow bottlenecks, the spread of possible allocations is much wider for varying the activation ratio (see left panel of *Figure 3A*) compared to varying the density ratio. Thus, for 2D sensory sheets, the activation ratio more strongly determines the possible allocation than does the density ratio. Specifically, this means that the allocation regime, that is, whether the allocation expands, contracts, or exhibits both behaviours across all bottleneck widths, is more dependent on relative receptor activation than densities (*Figure 3B*).

Finally, it is likely that regions with higher receptor densities will also show greater activation than lower density regions. For example, in touch, the regions on the hand with the highest receptor densities also are the most likely to make contact with objects (*Gonzalez et al., 2014*). In these cases, both effects reinforce each other and drive the resulting allocation further from proportional allocation (see orange lines in *Figure 3A* and green ellipse in *Figure 3B*).

## Resource limits determine the strength of plasticity under changes in stimulus statistics

Over most of the lifetime of an organism, the resources available for processing sensory information and the density of sensory receptors should be relatively constant. The stimulus statistics, on the other hand, can and will change, for example, when encountering a new environment or learning new skills. These changes in stimulus statistics should then affect sensory representations, mediated by a variety of plasticity mechanisms. For example, increased stimulation of a digit will lead to an expansion of that digit's representation in somatosensory cortex (*Jenkins et al., 1990*).

We asked how representations should adapt under the efficient coding framework and whether resource limits would affect the resulting changes. To answer this question, we calculated optimal allocations for different bottleneck widths, receptor densities, and stimulus statistics. We then introduced a change in stimulus statistics and re-calculated the resulting allocations (see illustration in *Figure 4A*). As expected, we found that when increasing the receptor activation over a region (e.g. by increasing

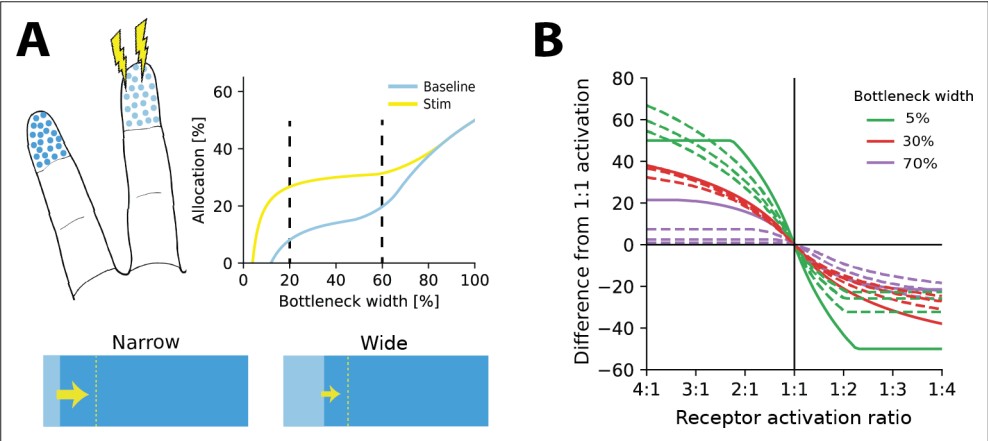

**Figure 4.** Re-allocation to account for changes in stimulus statistics. (**A**) Top left: illustration of problem setup. Increased stimulation is applied to the middle digit (yellow symbols), leading to changes in optimal allocations. Top right: optimal allocations for baseline (blue) and stimulation (yellow) conditions across all bottleneck widths. Stimulation of the middle finger increases its representation, but the relative magnitude of the effect depends on the bottleneck width. Bottom: changes in allocation of the middle digit for two bottleneck widths (indicated by dashed lines above). The increase is proportionally larger for the narrow compared to the wide bottleneck. (**B**) Change in allocation when receptor activation for an input region increases (left half) or decreases (right half). Drastic changes in cortical allocation are observed for narrow bottlenecks (green lines), while wider bottlenecks (red and purple lines) induce only moderate change. Solid lines denote equal receptor density across both regions, while dashed lines show examples with varying density.

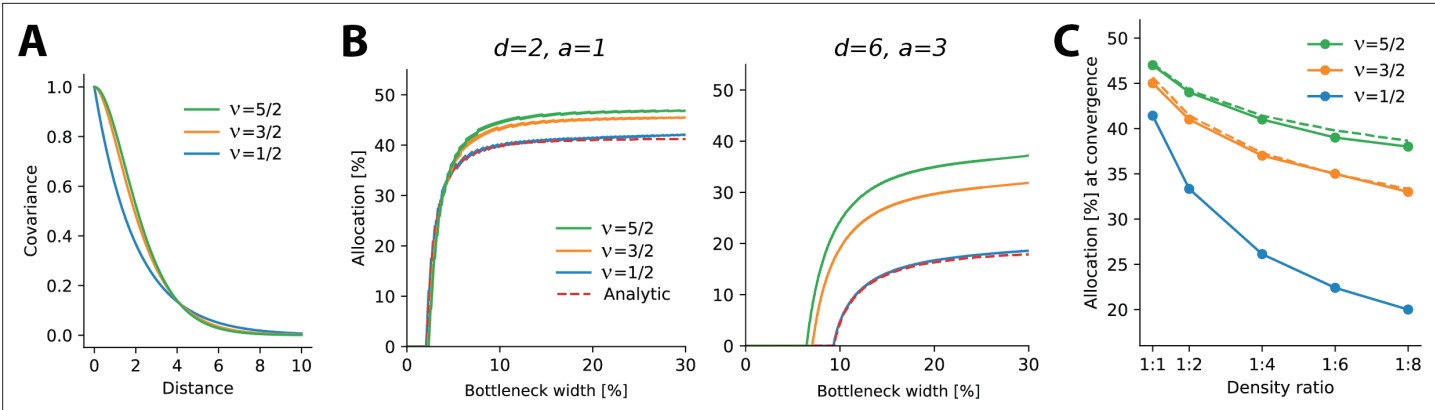

**Figure 5.** Allocations for other monotonically decreasing covariance functions. (**A**) Three covariance functions of different smoothness taken from the Matérn class, differing in the parameter $\nu$ (see 'Methods'). (**B**) Examples of numerically determined allocation for the covariance functions shown in (**A**) for two scenarios. Left: density ratio 1:2, equal activation. Right: density ratio 1:6, activation ratio 1:3. The red dashed line shows the analytic solution for the negative exponential covariance ($\nu = 1/2$). (**C**) Estimated allocations at convergence for different density ratios (horizontal axis) and a fixed activation ratio of 1:2. Solid lines denote numerical convergence, and dashed lines refer to fitted functions (see main text).

stimulation of the region), more neurons would be allocated to this region than to the other. Interestingly, however, this effect was dependent on the width of the bottleneck. The largest effects are seen for smaller bottlenecks and then diminish as the bottleneck size increases. *Figure 4B* demonstrates such allocation changes for three different bottleneck widths for a range of receptor densities and activation ratios. This suggests that plasticity should be relatively stronger under extreme resource constraints than in cases where limits on the information are generous.

## Generalization to other covariance functions

The results above relate to negative exponential covariance functions, where allocations can be solved analytically and it is unclear whether these findings generalize to other monotonically decreasing covariance functions and are therefore robust. Specifically, negative exponential covariances are relatively 'rough' and allocations might conceivably depend on the smoothness of the covariance function. To test directly how optimal allocations depend systematically on the smoothness of the covariance function, we numerically calculated allocations for three covariance functions of the Matérn class, whose smoothness depends on a single parameter $\nu$. The negative exponential considered so far is a special case of the Matérn class when $\nu = 1/2$ and we also considered $\nu = 3/2$ and $\nu = 5/2$ (see 'Methods'). As $\nu$ increases, the decay function becomes smoother, yielding larger correlations at short distances and smaller ones at farther distances (see *Figure 5A*).

We numerically calculated allocations for different density and activation ratios in 1D (*Figure 5B*). We focused on smaller bottleneck widths, where solutions are numerically stable, and observed a close match between the numerical and analytical solutions for the negative exponential covariance function (see dashed lines in *Figure 5B*).

This analysis yielded two main findings. First, allocation curves are qualitatively similar across all tested covariance functions in that regions with higher density, activation, or both are systematically over-represented. Furthermore, this effect is more extreme at smaller bottleneck widths. Second, the resulting allocations depend systematically on the smoothness of the covariance function, such that smoother covariances yield less extreme differences in allocation. This effect is most obvious when considering the convergence points at larger bottlenecks: smoother covariance functions induce a more uniform allocation (closer to a 50:50 split) than does the negative exponential (*Figure 5C*). We also noticed that the convergence points appeared to follow a simple function, namely, $\frac{1}{1+\sqrt[6]{ad}}$ for $\nu = 5/2$, $\frac{1}{1+\sqrt[4]{ad}}$ for $\nu = 3/2$ (see dashed lines in *Figure 5C*), and $\frac{1}{1+\sqrt{ad}}$ for $\nu = 1/2$ (which is the result obtained analytically, see Appendix 6).

In summary, monotonically decreasing covariance functions result in qualitatively similar allocations to the negative exponential studied earlier. However, precise allocations and convergence are

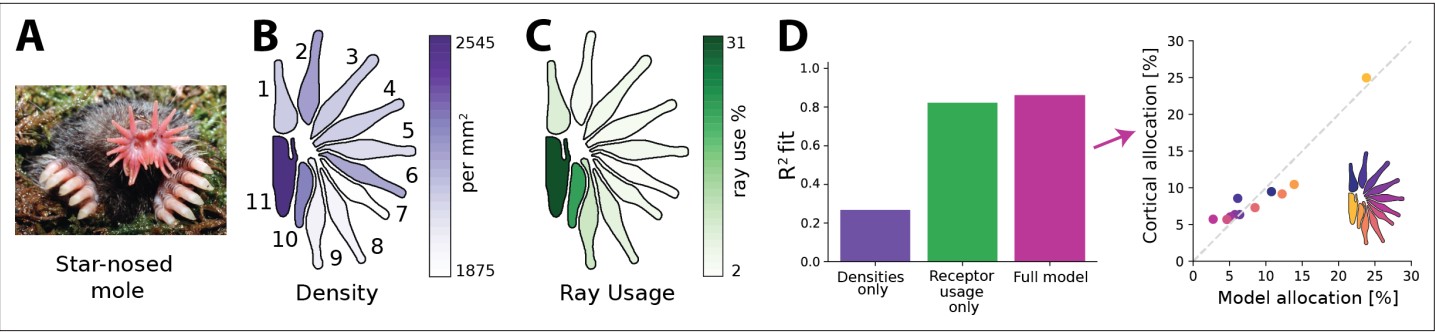

**Figure 6.** Resource allocation in the star-nosed mole. (**A**) Star-nosed moles have two sets of 11 tactile rays used for detecting and capturing prey. (**B**) Fibre innervation densities for each ray. (**C**) Typical usage percentages for each ray during foraging. Higher usage corresponds to greater average activation of receptors located on the corresponding ray. Typically prey is funnelled from distal rays towards ray 11, which is located next to the mouth. Ray outlines adapted from *Catania et al., 2011*. (**D**) Left: explained variance ($R^2$) between model predictions and cortical allocation for three different models: restricted to receptor density only (purple), restricted to receptor activation only (green), and a full model (pink) that incorporates accurate values for both factors. Results confirm previous findings that ray usage is a better predictor of cortical allocation than receptor densities alone. Additionally, we show that including both of these factors provides a marginal improvement to the fit, with the highest $R^2$ (86%). Right: predicted versus empirical cortical allocations for all rays. When including both density and activation parameters, the model provides a good fit to empirical measurements. Scatter plots for all models are available in *Figure 6—figure supplement 1B*. (**A**) reproduced from Figure 1C in *Catania et al., 2011*, copyright Kenneth Catania.

The online version of this article includes the following figure supplement(s) for figure 6:

**Figure supplement 1.** Additional figures for the star-nosed mole.

determined by the correlational structure: smoother covariance functions lead to less extreme differences in allocation.

## Predicting cortical magnification in the star-nosed mole

Finally, we investigated to what extent the procedure outlined in the previous sections might predict actual resource allocation in the brain. While our model is relatively simple (see 'Discussion'), decorrelation has been found to be a strong driver in early sensory processing, including in vision (*Atick and Redlich, 1992*; *Graham et al., 2006*; *Vinje and Gallant, 2000*), audition (*Clemens et al., 2011*; *Smith and Lewicki, 2006*), and touch (*Ly et al., 2012*), and one might therefore expect the approach to at least yield qualitatively valid predictions. As currently available empirical data makes it difficult to test the impact of different bottleneck widths on the resulting allocation directly (see 'Discussion'), we instead focused on another predicted outcome of the proposed model: the precise interaction between receptor density and receptor activation in driving resource allocation that we presented earlier. We picked the star-nosed mole as our model system because stimulus statistics, receptor densities, and cortical allocations have been precisely quantified. Moreover, the star-nosed mole displays considerable variation in all these parameters, presenting a good opportunity to put the model to the test.

The star-nosed mole is a mostly underground dwelling creature relying on active tactile sensing while foraging for prey (*Catania and Kaas, 1997*; *Catania, 2020*). This process is facilitated by two sets of 11 appendages arranged in a star-like pattern that make up the mole's nose (*Figure 6A*). Individual rays are tiled with tactile receptors known as Eimer's organs that detect prey, which is then funnelled towards the mouth (*Catania and Kaas, 1997*; *Sawyer and Catania, 2016*). The density of fibres innervating the Eimer's organs differs across the rays, with rays closer to the mouth exhibiting higher densities (*Figure 6B*). The rays also vary in size and location with respect to the mouth. This affects their usage as the rays closest to the mouth encounter tactile stimuli much more frequently than other rays (*Figure 6C*). In the cortex, a clear topographic representation of the rays can be found (*Catania and Kaas, 1997*). However, the extent of the cortical ray representations is not proportional to the physical size of the rays. Ray 11, which sits closest to the mouth, is cortically magnified several fold and is considered the tactile equivalent of the visual fovea (*Catania and Remple, 2004*).

Previous work by *Catania and Kaas, 1997* found that cortical sizes are correlated more strongly with the different patterns of activation across the rays, rather than their innervation densities. Using

the empirical data quantifying receptor densities and stimulus statistics from this study, we investigated whether the efficient coding model could predict typical cortical representation sizes for each ray (see 'Methods' for details) and whether innervation densities or usage would lead to more accurate allocation predictions. As the bottleneck size between periphery and cortex is unknown for the star-nosed mole, we calculated the optimal allocations over all possible bottleneck sizes. Using the model described above, we calculated allocations considering three different scenarios. The first, 'density only', included accurate afferent densities, but with activations set to the mean over all rays. The second model, 'usage only', included accurate activation ratios but mean receptor densities. Finally, the 'full model' included accurate values for both factors. We found that the empirical cortical representation sizes are most accurately predicted by the models that include receptor activation—the 'receptor usage only' and 'full models' (*Figure 6D*)—suggesting the star-nosed mole could be employing an efficient coding strategy based on decorrelation in the neural representation of somatosensory inputs.

## Discussion

We examined efficient population coding under limited numbers of output neurons in cases of non-uniform input receptor densities and stimulus statistics. Instead of focusing on the precise structure of the receptive field, we asked which coarse region of the sensory sheet the receptive field would fall on. We showed that the resulting allocations are nonlinear and depend crucially on the width of the bottleneck, rather than being proportional to the receptor densities or statistics. Specifically, narrow bottlenecks tend to favour expansion of a single region, whereas for larger bottlenecks, allocations converge to a constant ratio between the regions that is closer to a proportional representation. Whether, across all possible bottlenecks, allocations are always expanded, contracted, or show both expansion and contraction depends on the relative density and activation ratios, but receptor activation plays a bigger role. When allocation changes due to novel stimulus statistics, a larger fraction of output neurons will switch their receptive field to another region for narrow compared to wide bottlenecks. Finally, we demonstrated that in the star-nosed mole a model that includes both accurate innervation densities and contact statistics provides a better fit to the sizes of somatosensory cortical regions than considering each of these factors alone.

### Comparison with previous approaches

A key feature of efficient coding population models is the non-uniform allocation of output neurons, whereby stimuli occurring at higher probabilities are represented by a greater number of neurons. A common approach is to use Fisher Information as a proxy for mutual information, enabling the calculation of optimal output neuron density and tuning curve placement given a distribution of sensory stimuli (*Ganguli and Simoncelli, 2010*; *Ganguli and Simoncelli, 2014*; *Ganguli and Simoncelli, 2016*; *Yerxa et al., 2020*). However, these approaches maximize information between the output population and the stimulus distribution itself, such that allocating additional neurons to a given region of the input space will always lead to an increase in the overall amount of information represented. In contrast, our approach assumes a finite number of input receptors in each region. We are thus asking a different question than previous research: once information about a sensory stimulus has been captured in a limited population of receptors, what is the most efficient way of representing this information? This framing implies that once all information from a given region has been fully captured in the output population, our method does not allocate further neurons to that region. Therefore, this also places a limit on the total size of the output population as this cannot exceed the total number of input receptors.

There has also been prior work on how bottlenecks affect sensory representations, though mostly focused on how different levels of compression affect receptive field structure. For example, *Doi and Lewicki, 2014* predicted receptive fields of retinal ganglion cells at different eccentricities of the retina, which are subject to different convergence ratios. More recently, such direct effects on representations have also been studied in deep neural networks (*Lindsey et al., 2019*). Finally, some approaches employ a different cost function than the mean-squared reconstruction error inherent to the principal component analysis (PCA) method used here. For example, the Information Bottleneck method (*Tishby et al., 2000*; *Tishby and Zaslavsky, 2015*) aims to find a low-dimensional representation that

preserves information about a specific output variable, while compressing information available in the input. It is quite likely that the choice of cost function would affect the resulting allocations, a question that future research should pursue.

## Implications for sensory processing

The quantitative comparison of the model results with the cortical somatosensory representation in the star-nosed mole provided limited evidence that one of the model predictions—namely, that receptor density and activation statistics should jointly determine cortical organization—is borne out in a biological model system. Further direct tests of the theory are hampered by the lack of reliable quantitative empirical data. Nevertheless, the model makes a number of qualitative predictions that can be directly compared to available data or tested in future experiments.

First, there is additional evidence that both receptor density and stimulus statistics drive allocation in neural populations. Similar to the star-nosed model, the primate somatosensory system also exhibits non-uniform distributions of both stimulus statistics and receptor density. Both of these factors are also broadly correlated, for example, receptor densities are higher on the fingertips, which are also more likely to be in contact with objects (*Gonzalez et al., 2014*). Importantly, while receptor densities alone can explain some of the magnification observed in the cortical somatosensory homunculus, they cannot account for this effect fully (*Corniani and Saal, 2020*). Indeed, evidence from non-human primates shows that cortical magnification also critically depends on experience (*Xerri et al., 1996*; *Xerri et al., 1999*). Similar results have recently been obtained from the brainstem in mice (*Lehnert et al., 2021*). Empirically reported differences in sizes of receptive fields between the digit tips and the palm have also been recreated from typical contact across the hand during grasping (*Bhand et al., 2011*), suggesting an influence of statistics of hand use. Both mechanoreceptor densities (e.g. *Verendeev et al., 2015*) and hand use statistics (*Fragaszy and Crast, 2016*) differ across primates, forming the basis for a potential cross-species study. The model presented here demonstrates how both effects can be treated within a single framework driven by information maximization.

Second, it appears that allocation in sensory systems with severely constrained resources is qualitatively in agreement with model predictions. As our results demonstrated, magnification should be more extreme the tighter the bottleneck. The best characterized and most clearly established bottleneck in sensory processing is likely the optical nerve in vision. Given that the optic nerve serves as a narrow bottleneck (approximately 12–27%; assuming 0.71–1.54 million retinal ganglion cells with 80% Midget cells [*Curcio and Allen, 1990*; *Perry et al., 1984*] and 4.6 million cones [*Curcio et al., 1990*]), and that the fovea contains a much higher density of cone receptors than the periphery (*Wells-Gray et al., 2016*; *Curcio et al., 1990*), the model would predict a large over-representation of the fovea, agreeing with experimental observations (see *Edmondson et al., 2019*, for further qualitative evidence). In order to further test the proposed model, comparisons could be made between individuals to study variations within a population. Specifically, it would be expected that optic nerves containing a high number of fibres would devote proportionally fewer of them to the fovea than optic nerves containing smaller numbers of fibres (assuming equal receptor densities and numbers in the retina). These comparisons could be extended across species, taking advantage of the fact that photoreceptor densities and optic nerve fibre counts differ across many primate species (*Finlay et al., 2008*). Along these lines, recent computational work has shown that the amount of neural resources allocated to the optic nerve can be expected to affect the structure of the receptive field itself (*Lindsey et al., 2019*). Finally, the extent of cortical areas can be controlled by experimental interventions in animals (*Huffman et al., 1999*), which would constitute a direct manipulation of the bottleneck.

## Re-allocation during development, learning, and ageing

Changing receptor densities, stimulus statistics, or resource limits over the life span of an organism should lead to a dynamic re-allocation of the available resources. The most common case will be changes in stimulus statistics as both receptor densities and resources should be relatively stable. In such cases, representations should adapt to the new statistics. For example in touch, changing the nature of tactile inputs affects cortical representations (*Coq and Xerri, 1998*; *Merzenich and Jenkins, 1993*; *Xerri et al., 1996*). Increasing statistics of contact over a region typically leads to expansion of

that region in the cortex. Our method suggests that the precise level of expansion would be dependent on the bottleneck width with larger effects observed for narrower bottleneck sizes.

For the other cases, changes in fibre numbers during development and ageing might be interpreted as a change in resources. For example, the optic nerve undergoes a period of rapid fibre loss early during development (*Sefton et al., 1985*; *Provis et al., 1985*). Similarly, fibre counts in the optic nerve decrease during ageing (*Dolman et al., 1980*; *Jonas et al., 1990*; *Sandell and Peters, 2001*). In this case, the model would predict a decrease in the size of peripheral representation in the bottleneck compared to the fovea. It is also possible that the receptor densities themselves may change. In touch, ageing leads to reductions in the densities of receptors in older adults (*García-Piqueras et al., 2019*). In such cases, we have effectively increased the bottleneck width relative to our receptor population, which again should lead to the re-allocation of resources.

## Expansion and contraction along the sensory hierarchy

Magnification of specific sensory input regions can be observed throughout the sensory hierarchy, from the brainstem and thalamus up to cortical sensory areas. Often, the amount of expansion and contraction differs between areas. For example, the magnification of the fovea increases along the visual pathway from V1 to V4 (*Harvey and Dumoulin, 2006*). Which of these representations might be best addressed by the model presented here? The main component of the model is decorrelation, which has been shown to be a driving principle for efficient coding in settings where noise is low (*Chalk et al., 2018*). This is generally the case in low-level sensory processing, for example, in touch (*Goodwin and Wheat, 2004*). Our results might therefore best match early sensory processing up to perhaps low-level cortical representations. Beyond this, it is likely that noise will be much higher (*Shadlen and Newsome, 1998*) and for efficient codes to shift away from decorrelation (*Hermundstad et al., 2014*). Furthermore, while distinct bottlenecks, whether on the number of neurons or the amount of information, are common in low-level processing, it is less clear whether such restrictions constrain cortical processing. Whether and how such different regimes should affect neural allocations remains an open question.

## Perceptual consequences

Does the allocation of output neurons lead to testable perceptual consequences? While we do not model neurons' receptive fields directly, allocating more neurons to a given region would increase perceptual spatial acuity for that region. Indeed, cortical magnification and perceptual acuity are correlated in both vision (*Duncan and Boynton, 2003*) and touch (*Duncan and Boynton, 2007*). At the same time, the absolute limits on spatial acuity are determined by the density of receptors in each input region. A naive allocation scheme that assigns output neurons proportional to the density of receptors would therefore result in perceptual spatial acuity proportional to receptor distance. Instead, as our results have shown, the allocation should not be proportional in most cases. Specifically, for narrow bottlenecks we would expect relatively higher spatial acuity for regions with high receptor density than might be expected from a proportional allocation. Conversely, for wider bottlenecks this relationship should be reversed and spatial acuity should be better than expected for lower density regions. In agreement with these results, it has been found that in vision spatial resolution declines faster than expected with increasing eccentricity, suggesting a narrow bottleneck in the optic nerve (*Anderson et al., 1991*).

A second consequence is that spatial acuity should be better in regions with higher activation probability even when receptor densities are equal. Indeed, spatial discrimination in touch improves with training or even just passive stimulation (*Van Boven et al., 2000*; *Godde et al., 2000*), up to a limit that is presumably related to receptor density (*Wong et al., 2013*; *Peters et al., 2009*). Assuming a fixed resource limit, training may offer improvements to some digits to the detriment of others. Whether this is indeed the case has to our knowledge not yet been empirically tested.

Finally, previous work has shown that non-uniform tuning curves across a population will lead to characteristic biases in perceptual tasks (*Wei and Stocker, 2015*). While the original formulation assumed that this heterogeneous allocation of output neurons was driven by stimulus statistics alone, we have shown here that it can also be a consequence of receptor densities. Thus, perceptual biases might also be expected to arise from neural populations that efficiently represent sensory inputs sampled by non-uniform receptor populations.

## Limitations and future work

We considered simple linear models based on decorrelation and demonstrated that even such seemingly straightforward models exhibit surprising complexity in how they manage trade-offs in resource allocation under constraints. Specifically, we found that output neurons were not generally allocated proportionally to input neurons or according to some other fixed rule. It therefore stands to reason that similarly complex trade-offs would manifest in more complex models, even though the precise allocations might differ. Nevertheless, since PCA is widely employed for decorrelation and dimensionality reduction, and therefore incorporated into many other algorithms, our results immediately generalize to several other methods. For example, it is straightforward to extend the model with additional constraints (e.g. response sparsity or power constraints) that would not affect the resulting allocations (see 'Methods' for details), and therefore the model presented here already covers a number of related models. This includes independent component analysis, which considers higher-order rather than second-order statistics, but relies on a whitened signal, which in the undercomplete (bottleneck) case is obtained via PCA (*Hyvärinen and Oja, 2000*). Similarly, some models that do incorporate sensory noise and maximize reconstruction accuracy also use an undercomplete set of principal components to reduce the dimensionality of the sensory signal (*Doi and Lewicki, 2014*). In both of these examples, the resulting receptive field structure will differ, but their allocation—where on the sensory sheet they will fall—will be governed by the same principles described earlier. However, we did not consider nonlinear models and previous work has demonstrated that response nonlinearities can make important contributions to decorrelation (*Pitkow and Meister, 2012*). Additionally, the precise structure and origin of noise in nonlinear models have been demonstrated to affect efficient coding strategies (*Brinkman et al., 2016*) and might therefore also influence the resulting allocations. Future work should explore resource allocation in such complex models.

We considered allocations for monotonically decreasing covariance functions. Choosing a negative exponential function and assuming that activations are uncorrelated across different input regions allowed us to derive an analytical solution. How justified are these assumptions? Many sensory systems will likely obey the intuitive notion that receptor correlations decrease with distance; however, there are notable exceptions, for example, in the auditory system (*Terashima and Okada, 2012*). In touch, the sensory system we are mainly concerned with here, contact might often be made with several fingers (for humans) or rays (for the star-nosed mole) simultaneously, which would induce far-ranging non-monotonic correlations. Unfortunately there is little quantified evidence on the strength of these correlations, rendering their importance unclear. At least for the star-nosed mole, their prey is often small compared to the size of the rays, which move mostly independently, so cross-ray correlations might be low. Furthermore, the receptive fields of neurons in early somatosensory cortex of both humans and star-nosed moles are strongly localized to a single appendage and lie within cortical sub-regions that are clearly delineated from others (*Sur et al., 1980*; *Nelson et al., 1980*; *Catania and Kaas, 1995*). If long-range non-monotonic correlations were strong, we would expect to find many multi-appendage receptive fields and blurry region boundaries. As this is not the case, it therefore stands to reason that either these correlations are not very strong or that there is some other process, perhaps during development, that prevents these correlations affecting the final allocation. Either way, our assumption of monotonically decreasing covariance functions appears to be a good first-order match. Still, the question of how to arrive at robust allocations for more complex covariance functions is an important one that should be considered in future research. While it is possible in principle to solve the allocation problem numerically for arbitrary covariance functions, in practice we noticed that the presence of small numerical errors can affect the sorting process and caution is therefore warranted, especially when considering non-monotonic functions.

## Methods

Our main goal was to derive a method for efficiently allocating output neurons to one of several input regions with different correlational response structure in the presence of constraints on the number of output neurons or amount of information being transmitted. In the following, we focus on the main rationale and equations, while specific proofs can be found in Appendix 1–6. First, we outline the framework for combined whitening and dimensionality reduction that is employed. Next, we demonstrate how this framework can be applied to multiple input regions with different statistics and

densities of receptors, and how calculation of the eigenvalues of region-specific covariance matrices solves the problem of resource allocation. Finally, we demonstrate how the problem can be solved analytically for a certain choice of covariance function. The Python code implementing the equations that can be used to recreate the figures in this article is available on GitHub (*Edmondson, 2021*).

## Combined whitening and dimensionality reduction

We assume that receptors are arranged on a 2D sensory sheet. Correlations in the inputs occur as receptors that are nearby in space have more similar responses. Restricting ourselves to such second-order statistics and assuming that noise is negligible, information is maximized in such a setup by decorrelating the sensory inputs. Here, we decorrelate using a simple linear model. To model the bottleneck, we restrict the number of outputs to $m < n$, where $n$ is the total number of receptors.

If the inputs are represented as a matrix $\boldsymbol{X}$ of dimensions $n \times z$ (where $z$ is the number of sensory input patterns), then our goal is to find an $m \times n$ dimensional matrix $\boldsymbol{W}$ such that $\boldsymbol{WX}$ is uncorrelated:

$$\boldsymbol{X}^T\boldsymbol{W}^T\boldsymbol{W}\boldsymbol{X} = \boldsymbol{I}. \tag{5}$$

This is achieved by setting $\boldsymbol{W} = \boldsymbol{\Sigma}^{-\frac{1}{2}}$, where $\boldsymbol{\Sigma} = \boldsymbol{X}^T\boldsymbol{X}$. Solutions can then be expressed in terms of the diagonal matrix of eigenvalues, $\boldsymbol{\Lambda}$, and eigenvectors, $\Phi$, of the covariance matrix $\boldsymbol{\Sigma}$:

$$\boldsymbol{W} = \boldsymbol{P}\boldsymbol{\Lambda}^{-\frac{1}{2}}\Phi^T. \tag{6}$$

Whitening filters are not uniquely determined and optimal decorrelation is obtained with any orthogonal matrix $\boldsymbol{P}$. Setting $\boldsymbol{P} = \boldsymbol{I}$ (plain whitening/PCA) leads to the form shown in the main results section (*Equation 1*). Localized receptive fields are obtained by setting $\boldsymbol{P} = \Phi$, which is known as zero-phase component analysis. In cases with a bottleneck, the solution involves solving an Orthogonal Procrustes problem (*Doi and Lewicki, 2014*) to find $\boldsymbol{P}^*$, an $m$-dimensional orthogonal matrix (where $m$ is the size of the bottleneck) which minimizes the reconstruction error of the inputs and a set of ideal local receptive fields $\boldsymbol{W}_{opt}$:

$$\boldsymbol{P}^* = \min_{\boldsymbol{P}} \left\| \boldsymbol{W}_{opt} - \boldsymbol{P}\boldsymbol{\Lambda}^{-\frac{1}{2}}\Phi^T \right\|_F^2, \tag{7}$$

where $\|\cdot\|_F$ denotes the Frobenius norm, and $\boldsymbol{\Lambda}$ and $\Phi$ are as above but retaining only those components with the $m$ largest eigenvalues. As the optimally whitened solution is independent of $\boldsymbol{P}$, additional constraints on the solution can be enforced. For example, power constraints are often added, which either limit the total variance or equalize the variance across output neurons, and additional sparsity constraints can be placed on the output neuron's activity or their receptive field weights (*Doi et al., 2012*; *Doi and Lewicki, 2014*). For example, to optimize the sparsity of the weight matrix, one would define an appropriate cost function (such as the $L^1$ norm of the weight matrix), then iteratively calculate the gradient with respect to $\boldsymbol{W}$, take an update step to arrive at a new $\boldsymbol{W}_{opt}$, and determine $\boldsymbol{P}$ as described in *Equation 7* (see *Doi and Lewicki, 2014*, for further details). Importantly for our problem, such additional constraints will affect the precise receptive field structure (through $\boldsymbol{P}$), but not the eigenvalues and eigenvectors included in $\boldsymbol{\Lambda}$ and $\Phi$, respectively. As we will see in the following section, our solution relies on the eigenvalues only, and we can therefore solve the allocation problem irrespective of the precise receptive field structure or additional constraints on the solution.

## Extension to multiple input regions

For our specific problem, we are interested in the case of multiple input regions with different correlational structure (i.e. due to differing receptor density or activation). To simplify the derivations, we approximate different input regions as independent, such that the overall covariance matrix will be a block diagonal matrix. The covariance $\boldsymbol{\Sigma}$ for two input regions, $R_1$ and $R_2$, can then be expressed as follows:

$$\boldsymbol{\Sigma} = \begin{bmatrix} \boldsymbol{\Sigma}^{(R1)} & \boldsymbol{0} \\ \boldsymbol{0} & \boldsymbol{\Sigma}^{(R2)} \end{bmatrix} \tag{8}$$

This assumption turns out to be a reasonable approximation when region sizes are relatively big and correlations typically do not extend far across the sensory sheet (see *Edmondson et al., 2019*, for a comparison between block and non-block region covariance matrices in 1D). Furthermore, in many sensory systems, the borders between regions of differing density tend to be relatively narrow. For example, in touch, the digits of the hand are spatially separated, and regions of differing densities, for example, between the digit tips and proximal phalanges, neighbour along the short rather than long axis of the digit. In the star-nosed mole, rays of different innervation densities are separated and neighbour only along their connection to the rest of the nose. However, the block matrix approximation might be problematic in cases with many very small adjacent regions with strong, far-ranging correlations.

The eigenvalues and eigenvectors of a block diagonal covariance matrix also follow the block diagonal form and can be calculated from the individual region covariances alone by a simple application of the Cauchy interlacing theorem. Thus, the corresponding eigenvalues and eigenvectors are

$$\mathbf{\Lambda} = \begin{bmatrix} \mathbf{\Lambda}^{(R1)} & \mathbf{0} \\ \mathbf{0} & \mathbf{\Lambda}^{(R2)} \end{bmatrix} \quad \text{and} \quad \Phi = \begin{bmatrix} \Phi^{(R1)} & \mathbf{0} \\ \mathbf{0} & \Phi^{(R2)} \end{bmatrix}. \tag{9}$$

Due to the imposed bottleneck, only the $m$ largest eigenvalues from the combined set $\mathbf{\Lambda}$ will be retained. If receptive fields are localized such that they are constrained to fall within a single input region, then an eigenvalue selected from $\mathbf{\Lambda}^{(R1)}$ indicates that the receptive field of the corresponding output neuron will fall onto region $R_1$, and analogously for $R_2$. This fact holds independent of the structure of $P$ that is chosen in *Equation 7* because in order to preserve decorrelation a given region cannot contain more output neurons than eigenvalues retained from this region. In the following, we show how the eigenvalues can be calculated analytically for certain covariance functions.

## Calculation of eigenvalues for negative exponential covariance functions

We model the covariance between receptors as a negative exponential function. The covariance matrix is then calculated as a function of distance between pairs of receptors (see *Figure 1C*). We will first go through the calculation of eigenvalues for the baseline region $R_1$, and then continue with the derivation for $R_2$, which exhibits different receptor density, receptor activation, or both.

For region $R_1$ the covariance between receptors is calculated as

$$\mathbf{\Sigma}_{ij}^{(R1)} = e^{-\gamma |x_i - x_j|}, \tag{10}$$

where $x_i$ and $x_j$ are the locations of the $i$th and $j$th receptors, and $\gamma$ is the decay constant.

The corresponding eigenvalues for an exponential covariance function in the continuous domain can be calculated analytically. The eigenvalue–eigenvector problem is expressed as an integral homogeneous equation, such that for $R_1$ we get

$$\lambda_k \phi_k(x) = \int_0^L e^{-\gamma |x - y|} \phi_k(y) dy, \tag{11}$$

where $\phi_k(x)$ is the $k$th eigenfunction and $\lambda_k$ its corresponding eigenvalue. The domain length $L$ is the input region size for one of the dimensions.

It can be shown that solutions to this problem can be related to the Laplacian operator (see Appendix 2 and Appendix 3 for proofs), such that

$$\lambda_k = \frac{2\gamma}{\mu_k + \gamma^2}, \tag{12}$$

where $\mu_k$ are the eigenvalues of the Laplacian operator.

The general solution for the Laplacian eigenvalue problem for a 2D rectangle with Dirichlet boundary conditions is (*Strauss, 2007*)

$$\mu_k = \mu_{l,m} = \frac{l^2 \pi^2}{L_1^2} + \frac{m^2 \pi^2}{L_2^2}, l, m = 1, 2, ..., \tag{13}$$

where $L_1$ and $L_2$ are the sizes of the domain for each dimension.

To calculate the covariance for $R_2$, we need to take into account the potentially different receptor density and response variance for this region. Denoting the ratio of the response variances between both regions by $a$, and the ratio of receptor densities by $d$, the covariance for $R_2$ can be expressed as

$$\mathbf{\Sigma}_{ij}^{(R2)} = ae^{-d\gamma|x_i - x_j|}. \tag{14}$$

It can be seen that $a$ scales the overall covariance matrix (see also **Figure 1C**), while $d$ changes the spatial extent of the correlations and thereby implicitly accounts for the different receptor density.

The calculation of the eigenvalues for $R_2$ proceeds analogously. The eigenvalue–eigenvector problem is given as

$$\lambda_k \phi_k(x) = \int_0^{L\sqrt{d}} ae^{-d\gamma|x - y|} \phi_k(y) dy \tag{15}$$

Since the receptor density ratio $d$ causes an implicit stretching of the space for the higher density region, the region length $L$ needs to be adjusted in order to keep the effective size of the region constant. In 2D, each axis is therefore scaled by $\sqrt{d}$, resulting in an upper integration limit of $L\sqrt{d}$.

## Allocation in the bottleneck

Given a sensory system with limited representational capacity, different regions may be allocated different amounts of resources. Here, we calculate the allocations over different bottleneck widths for two regions, while the extension to multiple regions is given in Appendix 5. In the following, we assume 2D square regions of equal size for ease of analysis (see Appendix 6 for the equivalent solution in 1D). A single variable $L$ is therefore used to denote the lengths of the squares. Following **Equation 12**, the eigenvalues for regions $R_1$ and $R_2$ can now be calculated as

$$R_1: \quad \lambda_{l,m}^{(R1)} = \frac{2\gamma}{\pi^2 L^{-2}(l^2 + m^2) + \gamma^2} \tag{16}$$

$$R_2: \quad \lambda_{n,o}^{(R2)} = \frac{2\gamma a\sqrt{d}}{\pi^2 L^{-2}(n^2 + o^2) + \gamma^2} \tag{17}$$

where $l, m$ and $n, o \in \mathbb{N}$ enumerate different eigenvalues for regions $R_1$ and $R_2$, respectively.

In order to calculate how many output neurons are allocated to $R_1$ and $R_2$ for different bottleneck widths, we will need to establish an ordering of the eigenvalues, such that for each pair $(l, m)$ we can determine the sorted rank of the eigenvalues. In contrast to the 1D case (see Appendix 6), there is no natural ordering of the eigenvalues in two dimensions; however, a close approximation can be obtained by calculating the number of lattice points enclosed by a quarter circle with radius $p = l^2 + m^2$ (see Appendix 4 for full details). Denoting this function as $N(p)$ and setting $p^{(R1)} = l^2 + m^2$ and $p^{(R2)} = n^2 + o^2$, we can then calculate the number of eigenvalues allocated to $R_1$ as a function of the number of neurons allocated to $R_2$, by setting $\lambda^{(R1)} = \lambda^{(R2)}$ and solving for $p^{(R2)}$. This yields

$$p^{(R2)} = a\sqrt{d}p^{(R1)} + \frac{L^2\gamma^2 a\sqrt{d} - L^2\gamma^2}{\pi^2}. \tag{18}$$

As we allocate more neurons to region $R_1$, the ratio $\frac{N(p^{(R1)})}{N(p^{(R2)})}$ simplifies to $\lim_{R_1 \to \infty} \frac{N(p^{(R1)})}{N(p^{(R2)})} = a\sqrt{d}$. The fraction of neurons allocated to each region therefore depends on the size of the bottleneck and converges to $\frac{1}{1+a\sqrt{d}}$ and $\frac{a\sqrt{d}}{1+a\sqrt{d}}$ for $R_1$ and $R_2$, respectively.

## Alternative covariance functions

To test generalization to other covariance functions that decrease monotonically with receptor distance, we tested a number of functions from the Matérn class, in which the parameter $\nu$ controls the smoothness of the function. The negative exponential covariance function we employed in previous sections is equivalent to a Matérn function with $\nu = 1/2$. Larger values of $\nu$ lead to progressively higher correlations for smaller distances and lower correlations for larger distances. Specifically, we tested a Matérn function with $\nu = 3/2$:

$$\mathbf{\Sigma}_{ij}^{\nu=3/2} = \left(1 + \gamma\sqrt{3}|x_i - x_j|\right) e^{-\gamma\sqrt{3}|x_i - x_j|}, \tag{19}$$

and with $\nu = 52$:

$$\Sigma_{ij}^{\nu=5/2} = \left(1 + \gamma\sqrt{5}|x_i - x_j| + \frac{\gamma^2 5|x_i - x_j|^2}{3}\right) e^{-\gamma\sqrt{5}|x_i - x_j|} \tag{20}$$

For simplicity and to obtain numerically stable solutions, all calculations were performed for 1D regions only. For all cases tests, the baseline region contained 500 receptors, while the number of receptors for the other region was determined by the density ratio. Eigenvalues were calculated from the covariance matrix using the *eigh* method of the *numpy* Python package (*Harris et al., 2006*). Comparing the numerically obtained allocation for the negative exponential covariance function with its analytical solutions showed a close match for most bottleneck widths. However, numerical errors increased for wider bottlenecks, where eigenvalues became very small and their decay flattened out, affecting the sorting process. For the analyses here, we therefore restricted ourselves to bottleneck widths where a close match between the numerical and analytical solutions could be obtained. This range was sufficient to demonstrate allocation at narrow bottlenecks and estimate the convergence point for larger ones.

## Calculations for star-nosed mole

The 11 rays were approximated as 2D square regions with areas set to their reported sizes (*Sawyer and Catania, 2016*). Receptor densities for each ray were calculated as the sensory fibre innervation per mm² (*Catania and Kaas, 1997*). Approximations of receptor activation on each ray were calculated from empirical data of prey foraging interactions recorded by *Catania and Kaas, 1997*. Contact statistics were converted to receptor activation probabilities with receptors following a Bernoulli distribution. Finally, activation variance was calculated as the variance of the Bernoulli distribution (see Appendix 1). The decay rate γ of the negative exponential covariance function was determined for each ray using a model of the typical extent of receptor activation during interaction with prey stimuli of varying sizes. Each ray interacts with varying prey sizes at different frequencies. For example, ray 11 is typically contacted by smaller stimuli more often than other rays. A 2D model of the rays was used to simulate average responses to each stimulus size. Each model ray was tiled with receptors, and circular stimuli of different sizes were then randomly placed over the ray. The radii and frequencies of each stimulus size were based on the prey model (*Catania and Kaas, 1997*). A ray receptor was marked as active if its coordinate position was within the bounds of the stimuli. Response covariance between receptors was then calculated and an exponential function was fit to find the γ decay parameter. See *Table 1* for the full set of parameters. The code implementing the receptor model is available on GitHub (*Edmondson, 2021*). To determine allocations, eigenvalues associated with each ray were calculated analytically, resulting in allocations for each ray at all bottleneck widths. Three models were compared: first, a 'density-only' model, which includes accurate receptor density ratios, but receptor activation ratio remains uniform across all rays; second, an 'activation-only' model, which includes heterogeneous receptor activation ratios, but uniform receptor density ratios across all rays; finally, the 'full model' combines both accurate densities and receptor activation ratios. Model allocations for each ray were compared to the cortical allocation empirical data from *Catania and Kaas, 1997*. As the bottleneck size for the star-nosed mole is unknown, the root-mean-square error (RMSE) was calculated for each model at all bottleneck widths. The bottleneck resulting in the lowest error was then selected for each. Allocations for rays at each bottleneck can be found in *Figure 6— figure supplement 1A*. For the 'activation-only' and 'full' models, the lowest RMSE values were for bottleneck widths of between 37 and 45%; for the 'density-only' model, the RMSE was similar over all bottlenecks widths (see *Figure 6—figure supplement 1C*). We also tested a 'baseline model' where

**Table 1.** Model parameters for the star-nosed mole.

| Ray | 1 | 2 | 3 | 4 | 5 | 6 | 7 | 8 | 9 | 10 | 11 |
|---|---|---|---|---|---|---|---|---|---|---|---|
| Size | 1.14 | 1.21 | 1.21 | 1.17 | 1.08 | 1.02 | 1.00 | 1.13 | 1.05 | 0.87 | 1.10 |
| Density | 45.78 | 47.14 | 45.82 | 45.69 | 45.2 | 46.91 | 43.3 | 44.01 | 44.26 | 47.92 | 50.46 |
| Activation | 0.03 | 0.01 | 0.02 | 0.01 | 0.02 | 0.01 | 0.02 | 0.02 | 0.04 | 0.08 | 0.11 |
| γ | 0.99 | 1.00 | 1.02 | 1.01 | 1.00 | 1.01 | 0.99 | 1.04 | 1.10 | 1.16 | 1.27 |

densities and activation were randomly selected for each ray within the possible range of parameters. The aim was to determine how much explained variance within the cortical allocations was due to the selection of the best-fitting bottleneck and how much was due to the specific density and activation parameters. A total of 20 random models were run, and the average $R^2$ was –0.09 (see *Figure 6— figure supplement 1D*).

## Acknowledgements

This work was supported by the Wellcome Trust (209998/Z/17/Z) and the European Union Horizon 2020 programme as part of the Human Brain Project (HBP-SGA2, 785907).

## Additional information

### Funding

| Funder | Grant reference number | Author |
| --- | --- | --- |
| Wellcome Trust | 209998/Z/17/Z | Hannes P Saal |
| European Commission | HBP-SGA2 785907 | Alejandro Jiménez Rodríguez |

The funders had no role in study design, data collection and interpretation, or the decision to submit the work for publication. For the purpose of Open Access, the authors have applied a CC BY public copyright license to any Author Accepted Manuscript version arising from this submission.

### Author contributions

Laura R Edmondson, Conceptualization, Software, Formal analysis, Investigation, Visualization, Methodology, Writing – original draft, Writing – review and editing; Alejandro Jiménez Rodríguez, Formal analysis, Validation, Methodology, Writing – original draft, Writing – review and editing; Hannes P Saal, Conceptualization, Supervision, Methodology, Writing – review and editing

### Author ORCIDs

Laura R Edmondson  http://orcid.org/0000-0001-9886-1121
Hannes P Saal  http://orcid.org/0000-0002-7544-0196

### Decision letter and Author response

Decision letter https://doi.org/10.7554/eLife.70777.sa1
Author response https://doi.org/10.7554/eLife.70777.sa2

## Additional files

### Supplementary files
• Transparent reporting form

### Data availability

No data was generated for this study. All equations and model parameters are included in the manuscript and supporting files. Additionally, code implementing the model equations has been made available on Github at https://github.com/lauraredmondson/expansion_contraction_sensory_bottlenecks, (copy archived at swh:1:rev:dd6de7c05ae9443d034361b042b053b4f40717f5) (see also Methods section in manuscript).

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

# Appendix 1

## Stimulus statistics and response variance

Decorrelation works on second-order statistics and therefore stimulus statistics would only be taken into account by the model if they affect the covariance matrix. One way this can happen is through the extent of the spatial correlations (parameter γ in the covariance function). For example, in touch the size distribution of stimuli that would typically make contact with a given skin region might differ, leading to a different correlational structure. While we calculate allocations for different values of γ, we keep this value fixed across different input regions, for simplicity.

A more common case is that of receptors in one region being more active than receptors in another region. For example, in touch, the fingertips make contact with objects much more frequently than does the palm. An increased probability of making contact would translate into higher receptor response rates. In turn, higher response rates imply higher response variance, which would be directly reflected in the covariance matrix as a multiplicative scaling of the covariance function. For example, assuming that each receptor follows a simple Bernoulli distribution (either on or off, with a probability of $p$ being on), then the response variance can be calculated as $p(1 - p) = p - p^2$. Assuming that the likelihood of any receptor being active is generally low, the variance scales almost linearly with receptor activation. Differences in activation between two regions are represented by the activation ratio $a$ throughout the article.

## Appendix 2

### Relationship between PCA and Laplacian eigenvalue problem

#### Rationale

Let $\Omega$ be a region with a density of receptors $\rho$. In a 1D region $\Omega = [0, L]$, the density can be expressed as $\rho = \frac{N}{L}$ or number of receptors per unit length. Assuming an exponential decay of correlations, the covariance between receptors $i$ and $j$ is

$$C(i,j) = e^{-\gamma|i\Delta x - j\Delta x|}, \tag{21}$$

where $\Delta x = 1/\rho$ is the distance between receptors. Subsampling the space by taking a fraction $N/d$ of the original receptors, $d > 1$, the covariance for positions $i, j$, becomes

$$\tilde{C}(i,j) = e^{-\gamma|id\Delta x - jd\Delta x|} = e^{-d\gamma|i\Delta x - j\Delta x|}. \tag{22}$$

Therefore, we encode the changes in receptor density in a scaling of the exponential decay rate. For a given distribution of receptors, there is an induced partition of the interval $[0, L]$, therefore, for a fixed $x = i\Delta x$, the covariance in the $j$th bin is approximately equal to the area of the exponential covered in that bin:

$$\int_{(j-1)\Delta x}^{j\Delta x} C(x, y)dy \approx C(i\Delta x, j\Delta x)\Delta x,$$

Summing over all the bins, we arrive at the PCA problem:

$$\sum_{j=0}^{n} C(i\Delta x, j\Delta x)\phi(j\Delta x)\Delta x = \lambda\phi(i\Delta x). \tag{23}$$

The continuum limit is found formally as $\Delta x \to 0$.

#### Derivation

In order to find the optimal assignment for a given receptor density, we are interested in solutions to the following equation which can be seen as a continuous version of the traditional PCA problem with an exponentially decaying covariance matrix:

$$\lambda\phi(x) = \int_0^L e^{-\gamma|x-y|}\phi(y)dy, \tag{24}$$

where $\gamma$ is the decay rate. We are interested in solutions $\phi \in C^2(\mathbb{R})$, that is, twice differentiable solutions that satisfy appropriate boundary conditions.

**Theorem 1** *If $\phi$ is a solution of **Equation 24**, then it is an eigenfunction of the Laplacian operator with eigenvalues:*

$$\mu = \frac{2\gamma}{\lambda} - \gamma^2. \tag{25}$$

*That is, in one dimension, solutions $\phi$ satisfy*

$$-\frac{d^2}{dx^2}\phi(x) = \mu\phi(x). \tag{26}$$

*Proof*: Differentiating **Equation 24** twice using the Leibniz rule we obtain

$$\frac{d}{dx}\phi(x) \quad = \frac{\gamma}{\lambda}\left\{-\int_0^x e^{-\gamma(x-y)}\phi(y)dy + \int_x^L e^{\gamma(x-y)}\phi(y)dy\right\}, \quad \text{and} \tag{27}$$

$$\frac{d^2}{dx^2}\phi(x) \quad = \frac{\gamma}{\lambda}\left\{-2\phi(x) + \gamma\int_0^L e^{-\gamma|x-y|}\phi(y)dy\right\}. \tag{28}$$

The second term on the right-hand side can be replaced using **Equation 24** obtaining the desired result:

$$\frac{d^2}{dx^2}\phi(x) \quad = -\frac{2\gamma}{\lambda}\phi(x) + \gamma^2\phi(x), \quad \text{or} \tag{29}$$

$$-\frac{d^2}{dx^2}\phi(x) \;\;= \left(\frac{2\gamma}{\lambda} - \gamma^2\right)\phi(x). \tag{30}$$

The previous is a sufficient condition on the solutions to *Equation 24*. A necessary condition is given in the following theorem:

**Theorem 2** *A solution to Equation 26 is also solution to Equation 24 if it satisfies the following boundary conditions*:

$$\phi'(0) \;\;= \gamma\phi(0) \tag{31}$$

$$\phi'(L) \;\;= -\gamma\phi(L). \tag{32}$$

*Proof*: Assume $\phi$ is a solution to *Equation 26*. We proceed by convolving *Equation 26* on both sides with the kernel $e^{-\gamma x}$:

$$\int_0^x e^{-\gamma(x-y)}\phi''(y)dy = \mu\int_0^x e^{-\gamma(x-y)}\phi(y)dy. \tag{33}$$

Integrating by parts twice we get

$$\phi'(x) - e^{-\gamma x}\phi'(0) - \gamma\phi(x) + \gamma e^{-\gamma x}\phi(0) + \gamma^2\int_0^x e^{-\gamma(x-y)}\phi(y)dy = \mu\int_0^x e^{-\gamma(x-y)}\phi(y)dy. \tag{34}$$

Using *Equation 25* and *Equation 32*, we obtain

$$-\phi'(x) + \gamma\phi(x) = \frac{2\gamma}{\lambda}\int_0^x e^{-\gamma(x-y)}\phi(y)dy. \tag{35}$$

Repeating the procedure with the kernel $e^{\gamma x}$ in the interval $[x, L]$, yields

$$\phi'(x) + \gamma\phi(x) = \frac{2\gamma}{\lambda}\int_x^L e^{-\gamma(x-y)}\phi(y)dy. \tag{36}$$

Adding *Equation 35* and *Equation 36* we recover *Equation 24*, which finalizes the proof.

## Scaling

In some instances of our problem, the exponential covariance will be scaled by the activation ratio, *a*. In general, the same reasoning applies to any linear combination of solutions; therefore, our results extend to that case. In particular, we have the following result:

**Theorem 3** *The eigenvalues of the scaled covariance matrix, $C'(x, y) = aC(x, y)$ are*

$$\lambda_s = a\lambda, \tag{37}$$

*where $\lambda$ is an eigenvalue of the original problem.*

*Proof*: Let $\phi(x)$ be a solution of *Equation 24*. By linearity of the integral we have

$$\int_0^L ae^{-\gamma|x-y|}\phi(y)dy = a\int_0^L e^{-\gamma|x-y|}\phi(y)dy = a\lambda\phi(x) \tag{38}$$

## Appendix 3

### Solutions

In the previous section, we saw that solutions of the PCA problem (*Equation 24*) and the Laplacian eigenvalue problem (*Equation 26*) coincide if boundary conditions specified in *Equation 32* and *Equation 31* are met. Here, we show how these solutions relate to solutions of the boundary value problem of *Equation 26* with $\phi(0) = \phi(L) = 0$, which correspond to the eigenmodes of an idealized vibrating string fixed at the extremes. Such modes are considerably simpler than the exact ones and, as we show, are sufficient for our analysis.

### Eigenmodes of a vibrating string

Solving *Equation 26*, for $\kappa = \sqrt{\mu}, \mu > 0$, using standard methods (see *Simmons, 2016*, p. 355 onwards), we find the general expression for the eigenfunctions:

$$\phi(x) = A \sin(\kappa x) + B \cos(\kappa x), \tag{39}$$

with coefficients to be determined using the boundary conditions. The first boundary condition ($\phi(0) = 0$) implies that $B = 0$. The second boundary condition ($\phi(L) = 0$) gives the equation $\sin(\kappa L) = 0$, which is satisfied for $\kappa L = n\pi$, or

$$\mu = \frac{n^2 \pi^2}{L^2}, \tag{40}$$

where $n = 1, 2...$ is the index of the eigenvalue. The value of $A$ is arbitrary and is usually selected so the solution is normalized.

### Exact eigenvalues

In order to find the exact analytical eigenvalues of *Equation 24*, we again assume $\mu > 0$ and a solution of the form given in *Equation 39*. Using the first boundary condition (*Equation 31*, *Equation 32*) we get the following relationship:

$$A = \frac{\gamma}{\kappa} B, \tag{41}$$

and with the second boundary condition, we obtain

$$\tan \kappa L = \frac{\kappa A + \gamma B}{\kappa B - \gamma A}; \tag{42}$$

replacing *Equation 41*, we find the transcendental equation

$$\tan \kappa L = \frac{2\gamma\kappa}{\kappa^2 - \gamma^2}; \tag{43}$$

whose solutions lead to the exact eigenvalues of the continuous PCA problem.

### Relationship between exact and approximate eigenvalues

The main results presented in this study rely on the ordering of the eigenvalues only, rather than their precise magnitude. Any divergence between the exact and approximate eigenvalues is therefore relevant only if it affects this ordering. Furthermore, since allocation of eigenvalues to regions proceeds cumulatively as more eigenvalues are added, localized disturbances in a few eigenvalues will lead to only minor errors that are quickly corrected as more eigenvalues are added (since the correct allocation of eigenvalues to regions will be restored). In the following, we demonstrate that, comparing the ordering of the exact eigenvalues where the boundary conditions are derived from the integral equation (*Equation 43*) with that of the approximate ones (*Equation 40*), this order is altered only in a localized fashion that does not affect the analytical results of this article. To understand the changes in the ordering, consider the functions

$$f(\kappa) = \tan \kappa L, \tag{44}$$

and

$$g(\kappa) = \frac{2\gamma\kappa}{\kappa^2 - \gamma^2}, \tag{45}$$

whose intersection gives the solutions to *Equation 43*. The function $g$ has two hyperbolic-like branches, one to each side of the singularity $\kappa = \gamma$. We can distinguish two cases for the said intersection:

1. For $|\kappa| \gg \gamma$, $g$ approaches 0 and there is clearly only one intersection with $f$, which happens to be increasingly close to the approximate eigenvalues. Moreover, the function is also monotonic in this regime. This implies that the order is preserved.
2. For $|\kappa| \approx \gamma$, that is, close to the singularity, an additional root is inserted as $f$ cuts both branches of $g$ and the order is disturbed.

Altogether, this implies that, for the cases studied in this article that consider $\gamma < \pi$, only the first two eigenvalues might differ from the order given by the approximate solution. In general, in the intervals $(n\pi/2, (n+1)\pi/2)$ there will be one root for both equations except in the one closest to the singularity which will contain two. Moreover, from *Equation 25* we see that, for two regions with densities $\gamma$ and $d\gamma$ and Laplacian eigenvalues $\kappa_1$ and $\kappa_2$, $\lambda_1 > \lambda_2$ implies $\kappa_2 - d\kappa_1 > (1-d)d\gamma^2$. This inequality is satisfied by the exact eigenvalues, again, in all intervals except the one in which the singularity lies (as can be confirmed by expanding $f - g = 0$ around the approximate eigenvalues and solving for $\kappa$). We have compared the exact and approximate ordering for two regions for a number of (manageable) cases and have found the relationship described above to hold.

## Appendix 4

### Ordering in the 2D square case

For rectangle regions, the ordering can be solved by calculating the number of lattice points enclosed by a quarter ellipse (*Strauss, 2007*). Here, we use square regions and therefore the solution is the number of points enclosed in a quarter circle. The Gauss circle problem determines the number of integer lattice points which lie within a circle with radius $p \geq 0$, with its centre at the origin:

$$N(p) = \#\{(l, m) \in \mathbb{R} | l^2 + m^2 \leq p^2\}. \tag{46}$$

The number of lattice points within the circle is approximately equal to its area. The number of points within a square region can be approximated by calculating the area of the upper quarter of the circle (positive values only):

$$N(p) = \frac{\pi p^2}{4} \tag{47}$$

The number of eigenvalues in each region is therefore the area of the intersection of the circle and region.

For each region we calculate the number of lattice points enclosed by a quarter circle; for $R_1$ we set the radius equal to $l^2 + m^2$ and for $R_2$ to $n^2 + o^2$ (the solution of *Equation 18*), where $l, m, n, o = 1, 2...$ This number is approximated as the area of the quarter circle. For values of $l^2 + m^2$ or $n^2 + o^2$ greater than the total number of eigenvalues in each dimension ($L\sqrt{d}$), the approximation diverges from the true ordering as the area of the quarter circle becomes larger than the area of the lattice (region). In this case, a correction term is added:

$$N(p) = \begin{cases} \dfrac{\pi p}{4} - p \arccos\left(\dfrac{k}{\sqrt{p}}\right) - k\sqrt{p - k^2}, & \text{if } \dfrac{k}{\sqrt{p}} < 1. \\ \dfrac{\pi p}{4}, & \text{otherwise.} \end{cases} \tag{48}$$

where $p$ is either $l^2 + m^2$ or $n^2 + o^2$ for $R_1$ and $R_2$ respectively, and $k$ is the total number of eigenvalues in each region. Assuming a region size of $L$, with each receptor spaced one unit apart, $k = L^2$ for $R_1$, and $k = L^2 d$ for $R_2$.

## Appendix 5

### Allocation for multiple regions in 2D

For more than two regions, density and activation ratios for each additional region are calculated relative to a chosen baseline region. This leads to the following general form for calculation of the eigenvalues of any region $x$:

$$R_x: \ \lambda_{l,m} = \frac{2\gamma a \sqrt{d_x}}{l^2 \pi^2 L^{-2} \sqrt{d_b} + m^2 \pi^2 L^{-2} \sqrt{d_b} + \gamma^2 \sqrt{d_b}} \tag{49}$$

where $a$ is the region activation scaling ratio, $d_b$ is the density of the baseline region, and $d_x$ the density of region $x$. $l, m \in \mathbb{N}$ enumerate different eigenvalues for region $x$.

# Appendix 6

## Allocation for the 1D case

The 1D case for changes in density has previously been addressed in *Edmondson et al., 2019*. Here, we extend this to include changes in activation. For two regions $R_1$ and $R_2$, we can calculate their eigenvalues as

$$R_1: \quad \lambda_l^{(R1)} = \frac{2\gamma}{l^2\pi^2 L^{-2} + \gamma^2} \tag{50}$$

$$R_2: \quad \lambda_m^{(R2)} = \frac{2\gamma ad}{m^2\pi^2 L^{-2} + \gamma^2} \tag{51}$$

where $d$ is the ratio of higher and lower densities, $a$ is the ratio of receptor activation, $L$ is the length of the region, and $l, m \in \mathbb{N}$ denote successive eigenvalues for regions $R_1$ and $R_2$, respectively.

To calculate how many output neurons are allocated to region $R_2$ as a function of the number of neurons allocated to region $R_1$, we set $\lambda_l^{(R1)} = \lambda_m^{(R2)}$ and solve for $m$. This yields

$$m = \frac{\sqrt{ad(l^2\pi^2 + L^2\gamma^2) - L^2\gamma^2}}{\pi}. \tag{52}$$

It becomes apparent that for $l = 1$, that is, the first neuron allocated to region $R_1$, we have already assigned $m = \frac{\sqrt{ad(\pi^2 + L^2\gamma^2) - L^2\gamma^2}}{\pi}$ neurons to region $R_2$. As we allocate more neurons to region $R_1$, the ratio $\frac{m}{l}$ simplifies to: $\lim_{l\to\infty} \frac{m}{l} = \sqrt{ad}$. The fraction of neurons allocated to each region therefore depends on the size of the bottleneck and converges to $\frac{1}{1+\sqrt{ad}}$ and $\frac{\sqrt{ad}}{1+\sqrt{ad}}$ for $R_1$ and $R_2$, respectively.

