## [Editor Report]

The article develops a mathematical approach to study the allocation of cortical area to sensory representations in the presence of resource constraints. The theory is applied to study sensory representations in the somatosensory system. This problem is largely unexplored, the results are novel, and can be of interest to experimental and theoretical neuroscientists.

---

## [Decision Letter]

**Decision letter after peer review:**

Thank you for submitting your article "Expansion and contraction of resource allocation in sensory bottlenecks" for consideration by *eLife*. Your article has been reviewed by 3 peer reviewers, and the evaluation has been overseen by a Reviewing Editor and Michael Frank as the Senior Editor. The following individuals involved in review of your submission have agreed to reveal their identity: Memming Park (Reviewer #2).

Essential revisions:

While the reviewers found the work interesting and engaging, they agreed that some major revisions would greatly strengthen the manuscript. All have consulted with each other and the reviewing editor to compile these major prompts. Please also see the recommended, slightly more minor, revisions below as well as the public reviews provided by each reviewer.

1) Clarifying assumptions made

The paper should explicitly state the assumptions of the model, and discuss their consequences much earlier, and in more depth.

Currently, the Introduction and first paragraphs of the results equate efficient coding with decorrelation (e.g. paragraphs 3 and 4 of the introduction and opening sentences of the Results). The authors address the role of noise only briefly in the Discussion. This could be expanded. "Efficient coding" is a broad conceptual framework, and not a single algorithm such as decorrelation or even redundancy reduction. Currently, the manuscript makes the impression that the two are the same.

Other, unstated assumptions of the model (e.g. constraints on the total variance of the bottleneck, and the assumption that output activity is equalized across all neurons) should be made more explicit. How are those assumptions connected to biology? Is there evidence that cortical neurons in the somatosensory system are equally active? Even if there is not – this should be addressed explicitly.

Also, the analytical results are derived with a number of assumptions about the correlation structure of the stimuli and the structure of the bottleneck. Please clarify these assumptions as well. Importantly, please more clearly define the cost function. Is it simply the squared error loss between receptors and representation?

2) Putting the math more front-and-center

The fact that all results are derived analytically is a clear strength of the manuscript, but the authors have chosen to refrain from presenting any math in the Results section. This is very unfortunate, because this makes the formal arguments more fuzzy than need be. Please find a compromise between a technical and very high-level paper, and include the essential formal steps of the arguments in the main text.

3) Expanding the simulation examples provided, increasing the scope of the results

Along these lines, and looping back to prompt 1 and clarifying the assumptions in the model, please include one or two examples with more complicated input statistics or system architecture. For example, it is frequently the case that a narrow sensory bottleneck is followed by an expansion (e.g. retina -> optic nerve -> V1) – could the theory say something about resource allocation in such scenario?

In another example – what if the correlation structure is more complex? E.g. in the auditory nerve correlations between sensors can spatially non-local, due to the presence of harmonic sounds (e.g. Terashima and Okada, NeurIPS 2012). These are just suggestions of more complex scenarios which the authors could consider. These additional results do not have to be obtained analytically, and could be optimized through simulation. Such additions could greatly strengthen the current paper, broaden its scope, and increase its impact.

To show that those results are robust for small variations in the stimulus distribution, please provide numerical experiments with a different covariance function. Perhaps try a Matérn covariance function with nu=3/2 which is slightly smoother than the OU covariance. The experiments are numerically difficult, so this prompt only requires showing a case in a regime where things are numerically stable (this should be possible since the authors have done something similar in the NeurIPS 2019 paper). We would like to see that the presented results are not qualitatively very different.

4) Provide more details on the star-nosed mole data and predictions from the model

Please report in more detail how the predictions are generated for the data from the star-nosed mole. This could be done in the main text. For example, one could plot predictions for different sizes of the bottleneck. As a field, we should not be afraid of showing the range of predictions for different parameter values (e.g. for different bottleneck sizes), and demonstrating where they match the data, and where they do not – this does not necessarily weaken the theory. In this particular case – the theory could inform us about not-yet-measured properties of the system, such as the size of the bottleneck. Moreover, this will be more explicit way to report these results.

Furthermore, to model the stimulus distribution for star-nosed mole case, the authors simulated a noisy sensory system to convert contact statistics (assuming a circular object) to receptor activation statistics. Star-nose mole parameters (Appendix table) are in the code provided, but it doesn't seem like code for the conversion from the contact statistics to the receptor activation statistics is included. Please share this code, since the text in the manuscript is missing much of the necessary detail for the reproduction of this work.

Recommendations for the authors:

A) page 2 and Discussion: The authors discuss the use of Fisher Information and other measures for their goal. The use of Fisher Information for optimal population coding can be dangerous especially when taking N to infinity, as then the convergence conditions of 1/FI to the MSE are not met (see Berens et al. 2012). The crucial quantity for convergence is the SNR. Thus, the stated reason is not a reason not to use Fisher Information, although it's fine for the authors to choose not to use it. The logic should be improved in the text, though.

B) The claim that full model has better MSE is not very convincing. The difference between using density only and receptor only makes sense. But the full model has an additional degree of freedom, so naturally it should be better then either if tested on the same data that it was fit to. Unfortunately, the dataset the authors have is only 11 points, so there's no easy way of splitting them to show that this is effect is real. Hence, it's hard to make a strong claim about the tiny change in MSE. Please weaken the claim. Also, Figure 5D scatter plot for the full model is very similar to that of the receptor usage model (I ran your code). Please provide the corresponding plots explicitly in the supplement instead of the code only for transparency.

C) The model makes no distinction between the density of sensors in different regions and the exponential decay of the covariance functions. This is not a problem from the perspective of theoretical predictions, but perhaps it could be explained better (maybe with additional illustrations?).

D) It wasn't clear what the authors mean by "second-order". One assumes this means that the input statistics is Gaussian, but perhaps this is wrong. Did the authors mean that they are only analyzing the covariance functions, i.e. second-order statistics? Or was it that the neurons' response entropy were quantified by their variance? Please clarify.

E) In the text, 'activation' and 'activation ratio' are used interchangeably. It would be better to stick with the ratio since it's really only the relative activation of the second population that matters.

E) The 'σ' as 'decay constant' should be explicitly mentioned in the main text in addition to the methods section. At first it seems to be the length-scale, but it turned out to be inverse length-scale which was confusing, since σ is often used to denote the inverse length-scale. (Figures1C, 2C)

F) Please add the following citations to other theoretical work, which considered efficient encoding of somatosensory stimuli and the allocation of cortical space :

a) Bhand, M., Mudur, R., Suresh, B., Saxe, A., and Ng, A. (2011). Unsupervised learning models of primary cortical receptive fields and receptive field plasticity. Advances in neural information processing systems, 24, 1971-1979.

b) Plumbley, M. D. (1999). Do cortical maps adapt to optimize information density?. Network: Computation in Neural Systems, 10(1), 41.

G) page 2: In the discussion about bottlenecks in the retina, recent work by Lindsey et al. (2019, arxiv 1901:00945) should be cited.

H) page 5: "Decorrelation has been found to be a…" -> Please add citations to this claim

Figures:

I) Figure 1D: This figure is quite confusing. The decaying covariance is plotted for a 1-D case, the axes are unlabelled, and it takes a while to figure out that these two block matrices do form a single covariance matrix (even-though they correspond to different regions). Please revise this figure.

J) Figure 2D: small boxes with receptor location is misleading. It shows increased number of receptors when it's the activation ratio that's different. Please revise this figure.

K) Figure 3: A/B-labels seem to be too large; the meaning of the boxes of different colors and the lines was unclear to me. What determines e.g. the line length? The two panels in A should be clearly labeled with what quantity is varied.

L) Figure 4: What are the dashed lines? Is it correct that 4B is a simple restatement of the fact that the lines in 3A left are more spread for smaller bottlenecks? Interpreting this as a sign of more plasticity is an overstatement.

M) Figure 5: Please normalize the RMSE in Figure 5D with respect to the power of the signal. Otherwise the numbers aren't that meaningful, unless some clarifying point is missing. (Or, why not just report r^2?) Please also provide RMSE curves for all different bottleneck sizes and discuss how plausible the optimal values are. Given the right panel, it looks like the RMSE is mainly driven by ray 11, the outlier. How stable is the inferred conclusion if ray 11 is left out? Also, the authors seem to use RMSE to compare percentages. Could the authors comment on the suitability for that loss function?

[Editors' note: further revisions were suggested prior to acceptance, as described below.]

Thank you for resubmitting your work entitled "Expansion and contraction of resource allocation in sensory bottlenecks" for further consideration by *eLife*. Your revised article has been evaluated by Michael Frank (Senior Editor) and a Reviewing Editor.

The manuscript has been improved but there are some remaining issues that need to be addressed, as outlined below:

1. Regarding low-noise regime, it only appears in the discussion and it does not warn the readers that the numerous tiny tail eigenvalues are very sensitive to noise. This affects the robustness of the analysis and should be clearly stated in the text and appear after the first interpretation of the bottleneck results (Figure 2 and 2-supplement 4).

2. It is not stated clearly that decorrelation is an information maximizing transform only in absence of noise, which is a key assumption of the model. In the Methods (lines 493-494) it is written that adding the noise does not affect the eigendecomposition (and, in consequence, resource allocation), but if that is the case – it is an important result which should be highlighted and explored further – a sensory coding transformation which is independent of the noise level seems like an important finding. Please revise the text around this point.

3. It is misleading to claim that the analysis "maximizes information" without qualification in the abstract since this work only deals with variances. Please rephrase.

4. In Line 66, please add a qualifier to the statement that Fisher Information can be used to approximate Mutual Information. This is only true under certain assumptions in certain cases and specifically for optimal coding, the relationship is not straightforward (Berens et al. 2012, PNAS; Bethge et al. Neural Comp 2002).

5. lines 87-88: "We consider linear models which maximize information by decorrelating the sensory inputs, or, equivalently, minimize the mean-squared reconstruction error […]" – in a general case these two goals are not equivalent. The following citations referencing work by Doi and colleagues do not support this claim. This should be revised.

6. In Figure 1, it is hard to parse which colors are kept to mean the same thing across panels and which change meaning. An additional schematic of the model in this figure would help with clarity of the presentation.

7. In Equation 1, is P used at all? If not simplify. The explanation of P sounds complicated without really saying anything specific. Give an example of constraints that could be incorporated.

8. The assumption that the covariance between regions is zero is quite a strong one and seems central to the paper. This should be justified or at least discussed how realistic it is. In the finger tip example, one of course typically touches objects not with one finger but with multiple ones.

9. The brackets in Figure 3 were found to be very confusing, please revise.

10. The covariance functions in Figure 5 are not very different, it is not surprising that these result in very similar outcomes. Comment on this or consider revising.

---

## [Author Response]

Essential revisions:While the reviewers found the work interesting and engaging, they agreed that some major revisions would greatly strengthen the manuscript. All have consulted with each other and the reviewing editor to compile these prompts.

We would like to thank the reviewers and editors for their clear and insightful feedback. We have addressed the comments in a revised version. In brief, the major changes are as follows:

We have re-organised the manuscript as suggested, and clarified the model and its assumptions.Specifically, we now make clear that our results are not restricted to a particular decorrelation model but cover a wider class of models. This broadens the applicability of the findings, but also clarifies the essential assumptions the model rests on.We have included numerical results for additional covariance functions; these demonstrate that theresults hold qualitatively and are robust to small changes in the covariance structure, but also shed some light on how properties of the covariance structure affect the resulting allocation, which we believe will be insightful to the reader.Additional results have been provided for the star-nosed mole example and the language has beentoned down. Throughout the manuscript, many minor improvements have been made and some unwieldy sections of the mathematical appendix have been re-organised. Visualisation has also been improved.

Overall, we believe that the manuscript is much improved and think that the reviewers will agree.

1) Clarifying assumptions madeThe paper should explicitly state the assumptions of the model, and discuss their consequences much earlier, and in more depth.Currently, the Introduction and first paragraphs of the results equate efficient coding with decorrelation (e.g. paragraphs 3 and 4 of the introduction and opening sentences of the Results). The authors address the role of noise only briefly in the Discussion. This could be expanded. "Efficient coding" is a broad conceptual framework, and not a single algorithm such as decorrelation or even redundancy reduction. Currently, the manuscript makes the impression that the two are the same.

We have clarified the broader meaning of ‘efficient coding’ in the manuscript and now explain in more detail how decorrelation/redundancy reduction fits into this framework and why we have chosen to focus on it for this paper. We have also updated our discussion on the role of noise (specifically in the light of model assumptions, see next comment).

Other, unstated assumptions of the model (e.g. constraints on the total variance of the bottleneck, and the assumption that output activity is equalized across all neurons) should be made more explicit. How are those assumptions connected to biology? Is there evidence that cortical neurons in the somatosensory system are equally active? Even if there is not – this should be addressed explicitly.Also, the analytical results are derived with a number of assumptions about the correlation structure of the stimuli and the structure of the bottleneck. Please clarify these assumptions as well. Importantly, please more clearly define the cost function. Is it simply the squared error loss between receptors and representation?

Thank you for prompting us to clarify our assumptions. We realized that in our original submission, we did not make sufficiently clear which assumptions were strictly necessary for determining receptive field locations (i.e. allocation) and which were choices that need to be made in order to calculate precise receptive fields, but which do not affect the resulting allocation, and where therefore different choices might be made. This is now made clearer both at the start of the Results section (where we now present some key equations, see below) and in the Methods section.

In brief, the framework we employ (developed by Doi and Lewicki in several papers) allows calculation of linear receptive fields in the undercomplete case, enforcing decorrelation of the signals. Dimensionality reduction and decorrelation are achieved via PCA by calculating eigenvectors and eigenvalues. Since decorrelation is not unique, the additional degrees of freedom can be used to enforce additional constraints (via the matrix P in Equation 6). Doi and Lewicki showed how this could be done using an iterative procedure for optimizing P (Equation 7) for a number of constraints e.g. power constraints on individual or all output neurons, sparsity constraints, and even included some simple forms of noise. Crucially, the decorrelation part (eigenvectors) is unaffected by any of these constraints (as they only act through P), and it is this component that determines which region receptive fields will fall onto. Any additional constraints enforced through P will affect the shape and structure of the receptive fields, but not the region they fall onto. Additional constraints, such as power constraints or response sparsity, can thus be added to the model without affecting the results.

2) Putting the math more front-and-centerThe fact that all results are derived analytically is a clear strength of the manuscript, but the authors have chosen to refrain from presenting any math in the Results section. This is very unfortunate, because this makes the formal arguments more fuzzy than need be. Please find a compromise between a technical and very high-level paper, and include the essential formal steps of the arguments in the main text.

We now present the main equations associated with the efficient coding approach as well as the main steps of the derivation at the start of the Results section, aligned with the presentation of the problem and solution strategy presented in figure 1.

3) Expanding the simulation examples provided, increasing the scope of the resultsAlong these lines, and looping back to prompt 1 and clarifying the assumptions in the model, please include one or two examples with more complicated input statistics or system architecture. For example, it is frequently the case that a narrow sensory bottleneck is followed by an expansion (e.g. retina -> optic nerve -> V1) – could the theory say something about resource allocation in such scenario?

We have included numerical calculations for other covariance functions (see next comment). The model assumes a bottleneck, either as a limit on the number of neurons or on the information, and therefore will not directly be able to answer what the allocation should look like in an overcomplete scenario (such as the expansion from optic nerve to V1). The model might still apply if a subsequent stage places a clear limit on the amount of information rather than the number of neurons (see Figure2—figure supplement 4), however empirical data is lacking to answer such questions. We have expanded our discussion on this issue and now make clearer the limits and assumptions of the current model.

In another example – what if the correlation structure is more complex? E.g. in the auditory nerve correlations between sensors can spatially non-local, due to the presence of harmonic sounds (e.g. Terashima and Okada, NeurIPS 2012). These are just suggestions of more complex scenarios which the authors could consider. These additional results do not have to be obtained analytically, and could be optimized through simulation. Such additions could greatly strengthen the current paper, broaden its scope, and increase its impact.To show that those results are robust for small variations in the stimulus distribution, please provide numerical experiments with a different covariance function. Perhaps try a Matérn covariance function with nu=3/2 which is slightly smoother than the OU covariance. The experiments are numerically difficult, so this prompt only requires showing a case in a regime where things are numerically stable (this should be possible since the authors have done something similar in the NeurIPS 2019 paper). We would like to see that the presented results are not qualitatively very different.

We thank the reviewers for these suggestions, which have prompted us to test how well our findings generalize beyond negative exponential covariance functions. To do this we have focused on Matérn covariance functions, as suggested, of which the negative exponential can be considered a special case, while other instances exhibit a smoother covariance structure with higher correlation at small distances but smaller correlation at larger distances compared to the negative exponential. Therefore this class provides a natural starting point to test robustness and it turns out that the results prove rather insightful.

First, in numerical simulations, we found that our results held qualitatively across a range of Matérn parameter values. Allocations were more extreme at narrow compared to wide bottlenecks and regions with higher density and/or activations ended up over-represented. Thus, it appears the results obtained for negative exponential covariance functions do generalize to other monotonically decreasing covariance functions.

Second, we noted that precise allocations, and specifically the convergence points, differ across Matern functions and that they do so in a systematic fashion. Specifically, the smoother the covariance, the less extreme the over-representation. Moreover, the convergence points for the different Matérn functions appear to follow a simple mathematical function. Altogether these results suggest that the precise allocations for monotonically decreasing covariance functions are determined by their smoothness. We have included these findings as a new section in the Results (“Generalisation to other covariance functions”) and also added a new figure.

We also considered more complex covariance functions. Thank you for pointing out a specific example from the auditory system, which we weren’t aware of. Our approach requires that covariance functions decay with distance (to ensure that the approximation of the covariance matrix with a block matrix holds, which allows us to consider regions separately), however, they need not do so monotonically. We therefore tested a damped sinusoid covariance function numerically, whose shape resembles the complex harmonic covariance structure observed in the auditory system. Results suggested that in contrast to monotonically decreasing covariance functions, where allocations converge monotonically towards a limit, this more complex covariance function induced complex nonlinear allocations at different bottleneck widths. Unfortunately, it was difficult to obtain numerically stable solutions. These instabilities arise when the eigenvalue function flattens out and small numerical errors then affect the sorting of eigenvalues between the two regions. For monotonically decreasing covariance functions, this issue only arises close to convergence, however more complex covariance functions contain more such regions and we could not find a method to satisfactorily address this issue. We therefore elected not to include these results in the current manuscript, but have included additional text in the discussion instead.

4) Provide more details on the star-nosed mole data and predictions from the modelPlease report in more detail how the predictions are generated for the data from the star-nosed mole. This could be done in the main text. For example, one could plot predictions for different sizes of the bottleneck. As a field, we should not be afraid of showing the range of predictions for different parameter values (e.g. for different bottleneck sizes), and demonstrating where they match the data, and where they do not – this does not necessarily weaken the theory. In this particular case – the theory could inform us about not-yet-measured properties of the system, such as the size of the bottleneck. Moreover, this will be more explicit way to report these results.Furthermore, to model the stimulus distribution for star-nosed mole case, the authors simulated a noisy sensory system to convert contact statistics (assuming a circular object) to receptor activation statistics. Star-nose mole parameters (Appendix table) are in the code provided, but it doesn't seem like code for the conversion from the contact statistics to the receptor activation statistics is included. Please share this code, since the text in the manuscript is missing much of the necessary detail for the reproduction of this work.

We have included a supplementary figure with the predictions for all bottleneck sizes and have run some additional tests to verify that a good fit cannot be obtained by simply varying the bottleneck width if arbitrary densities and activation statistics are chosen. We have also clarified in the text the simulations of the decay parameter and calculation for bottleneck allocations. We have additionally switched from RMSE to R^2 and the results should now be easier to interpret. Finally, the code to convert the contact statistics and calculate the decay parameters for each ray has been added to the Github repository linked in the paper.

Recommendations for the authors:A) page 2 and Discussion: The authors discuss the use of Fisher Information and other measures for their goal. The use of Fisher Information for optimal population coding can be dangerous especially when taking N to infinity, as then the convergence conditions of 1/FI to the MSE are not met (see Berens et al. 2012). The crucial quantity for convergence is the SNR. Thus, the stated reason is not a reason not to use Fisher Information, although it's fine for the authors to choose not to use it. The logic should be improved in the text, though.

Thank you for the clarification. We have removed the reasoning about population sizes from the manuscript.

B) The claim that full model has better MSE is not very convincing. The difference between using density only and receptor only makes sense. But the full model has an additional degree of freedom, so naturally it should be better then either if tested on the same data that it was fit to. Unfortunately, the dataset the authors have is only 11 points, so there's no easy way of splitting them to show that this is effect is real. Hence, it's hard to make a strong claim about the tiny change in MSE. Please weaken the claim. Also, Figure 5D scatter plot for the full model is very similar to that of the receptor usage model (I ran your code). Please provide the corresponding plots explicitly in the supplement instead of the code only for transparency.

All models have the same number of variable parameters- density of each ray and activation. In the models where one of these is fixed, we set the fixed value to the mean of the values for all the rays. To test whether the range of bottleneck sizes results in the possibility to fit any given values for the rays, we included a further random simulation in the supplementary information. Here we randomise the values for the density and activation of each ray within the possible range of values for each. We find that with this randomisation of the values the model performs poorly on fitting even with a range of bottleneck sizes. This suggests that the model can only be fitted when using the empirically measured values.

C) The model makes no distinction between the density of sensors in different regions and the exponential decay of the covariance functions. This is not a problem from the perspective of theoretical predictions, but perhaps it could be explained better (maybe with additional illustrations?).

We have improved the text clarifying this relationship in the Results section. We now also make clear that in all examples shown receptors in the low-density region are spaced with a distance of 1, and thus the covariance decay is relative to this spacing.

D) It wasn't clear what the authors mean by "second-order". One assumes this means that the input statistics is Gaussian, but perhaps this is wrong. Did the authors mean that they are only analyzing the covariance functions, i.e. second-order statistics? Or was it that the neurons' response entropy were quantified by their variance? Please clarify.

Upon reflection, we have removed the term “second-order” from the text, as it was redundant and confusing. Instead, the term “decorrelation” that we already use implies that we are acting on second-order statistics. We don’t require the stimulus distribution itself to be Gaussian, or in fact the processing to only consider second-order statistics, but merely that decorrelation is a necessary step. If it is, this requires calculating the eigenvalues and eigenvectors of the covariance matrix and then the rest of our argument follows.

E) In the text, 'activation' and 'activation ratio' are used interchangeably. It would be better to stick with the ratio since it's really only the relative activation of the second population that matters.

We have clarified in the text what we mean by activation and activation ratio. Where we mean the activation ratio the text has been amended to clearly state this.

E) The 'σ' as 'decay constant' should be explicitly mentioned in the main text in addition to the methods section. At first it seems to be the length-scale, but it turned out to be inverse length-scale which was confusing, since σ is often used to denote the inverse length-scale. (Figures1C, 2C)

To avoid confusion, we have renamed the σ parameter to γ and have clarified its meaning in the text.

F) Please add the following citations to other theoretical work, which considered efficient encoding of somatosensory stimuli and the allocation of cortical space :a) Bhand, M., Mudur, R., Suresh, B., Saxe, A., and Ng, A. (2011). Unsupervised learning models of primary cortical receptive fields and receptive field plasticity. Advances in neural information processing systems, 24, 1971-1979.b) Plumbley, M. D. (1999). Do cortical maps adapt to optimize information density?. Network: Computation in Neural Systems, 10(1), 41.

Thank you for these additional references. We have added relevant points from these in the introduction relating to the optimal representation of information and magnification in cortical maps (Plumbley) and in the discussion on the influence of receptor density and statistics (Bhand).

G) page 2: In the discussion about bottlenecks in the retina, recent work by Lindsey et al. (2019, arxiv 1901:00945) should be cited.

This work was cited elsewhere in the paper, but indeed should also be mentioned in this context and we have now added the reference.

H) page 5: "Decorrelation has been found to be a…" -> Please add citations to this claim

We have added a number of citations to support this claim.

Figures:I) Figure 1D: This figure is quite confusing. The decaying covariance is plotted for a 1-D case, the axes are unlabelled, and it takes a while to figure out that these two block matrices do form a single covariance matrix (even-though they correspond to different regions). Please revise this figure.

We have updated this figure and believe the new version to be much clearer. We have switched to a 2D version for the covariance so that it aligns with the example introduced in the earlier panels. The fact that we are considering a single covariance matrix with block structure should also be more visible now.

J) Figure 2D: small boxes with receptor location is misleading. It shows increased number of receptors when it's the activation ratio that's different. Please revise this figure.

We have updated the figure and this should be more clearly indicated now.

K) Figure 3: A/B-labels seem to be too large; the meaning of the boxes of different colors and the lines was unclear to me. What determines e.g. the line length? The two panels in A should be clearly labeled with what quantity is varied.

Thank you for pointing out this issue. The original boxes and lines were supposed to indicate groupings of examples that showed similar behaviour. These indicators have been now replaced with symbols resembling brackets which should be more easily interpretable. We have also clarified the content of these plots with a fuller explanation in the figure caption. The size of the labels has been corrected.

L) Figure 4: What are the dashed lines? Is it correct that 4B is a simple restatement of the fact that the lines in 3A left are more spread for smaller bottlenecks? Interpreting this as a sign of more plasticity is an overstatement.

The dashed lines in panel A denote the two example bottlenecks for which allocations are visualized directly below. This has now been clarified in the figure legend.

M) Figure 5: Please normalize the RMSE in Figure 5D with respect to the power of the signal. Otherwise the numbers aren't that meaningful, unless some clarifying point is missing. (Or, why not just report r^2?) Please also provide RMSE curves for all different bottleneck sizes and discuss how plausible the optimal values are. Given the right panel, it looks like the RMSE is mainly driven by ray 11, the outlier. How stable is the inferred conclusion if ray 11 is left out? Also, the authors seem to use RMSE to compare percentages. Could the authors comment on the suitability for that loss function?

The plot in former figure 5D (now figure 6) has been changed to report the R^2 between allocations in model and cortical percentages from the empirical data at the best fitting bottleneck. We have provided the fits across a range of bottlenecks in figure 6—figure supplement 1.

[Editors' note: further revisions were suggested prior to acceptance, as described below.]

The manuscript has been improved but there are some remaining issues that need to be addressed, as outlined below:1. Regarding low-noise regime, it only appears in the discussion and it does not warn the readers that the numerous tiny tail eigenvalues are very sensitive to noise. This affects the robustness of the analysis and should be clearly stated in the text and appear after the first interpretation of the bottleneck results (Figure 2 and 2-supplement 4).

We have made the assumption of low noise and its implication clearer by making several changes to the text. In addition to the text in the discussion, we now make clear at the start of the Results section that we are assuming a low-noise regime. As requested, we have also added a note about the potential effects of noise after presenting the initial results in Figure 2:

“An additional consideration is the robustness of the results regarding small perturbations of the calculated eigenvalues. As allocation depends on the ranks of the eigenvalues only, low levels of noise are unlikely to affect the outcome for narrow bottlenecks, especially since eigenvalues are decaying rather steeply in this regime. On the other hand, allocation in wider bottlenecks is determined from small tail eigenvalues which are much more sensitive to noise (which is also evident when comparing the analytical solution to numerical ones). Allocation can therefore be expected to be somewhat less robust in those regimes.”

2. It is not stated clearly that decorrelation is an information maximizing transform only in absence of noise, which is a key assumption of the model. In the Methods (lines 493-494) it is written that adding the noise does not affect the eigendecomposition (and, in consequence, resource allocation), but if that is the case – it is an important result which should be highlighted and explored further – a sensory coding transformation which is independent of the noise level seems like an important finding. Please revise the text around this point.

We have edited the text in the Results section to clarify that decorrelation maximises information only in the absence of noise. We have removed the text in the Methods section regarding the effect of noise on the eigendecomposition. The idea was that under some specific noise models, receptive fields would be affected but not their location, thus rendering the resulting allocation the same. However, this issue is not the focus of the paper, not fully worked out and in any case has no bearing on the results of the paper, and removing it will avoid confusing the reader.

3. It is misleading to claim that the analysis "maximizes information" without qualification in the abstract since this work only deals with variances. Please rephrase.

We have removed the phrase ‘maximize information’ from the abstract:

“Building on work in efficient coding, we address this problem using linear models that optimally decorrelate the sensory signals.”

See also point 5 below, which concerns the wording in the main manuscript text.

4. In Line 66, please add a qualifier to the statement that Fisher Information can be used to approximate Mutual Information. This is only true under certain assumptions in certain cases and specifically for optimal coding, the relationship is not straightforward (Berens et al. 2012, PNAS; Bethge et al. Neural Comp 2002).

We have added a note to this section (with the suggested references) that explains this important caveat:

“In contrast to receptor density, there has been some work on how populations of neurons should encode non-uniform stimulus statistics using Fisher information (Ganguli and Simoncelli, 2010, 2014, 2016; Yerxa et al., 2020). This approach aims to approximate mutual information and is used to calculate optimal encoding in a neural population (Yarrow et al., 2012), however its use is restricted to specific conditions and assumptions (Berens et al., 2011; Bethge et al., 2002).”

5. lines 87-88: "We consider linear models which maximize information by decorrelating the sensory inputs, or, equivalently, minimize the mean-squared reconstruction error […]" – in a general case these two goals are not equivalent. The following citations referencing work by Doi and colleagues do not support this claim. This should be revised.

We have revised the wording in this section:

“We restrict ourselves to linear models and only consider second-order statistics of the sensory signals, such that redundancy reduction simplifies to decorrelation (see Hancock et al. 1992, Simoncell and Olshausen 2001, Doi and Lewicki 2005, for examples from the visual literature).”

We have also ensured that the wording in the Methods section is appropriate:

“We assume that receptors are arranged on a two-dimensional sensory sheet. Correlations in the inputs occur as receptors that are nearby in space have more similar responses. Restricting ourselves to such second-order statistics and assuming that noise is negligible, information is maximised in such a setup by decorrelating the sensory inputs. Here we decorrelate using a simple linear model.”

6. In Figure 1, it is hard to parse which colors are kept to mean the same thing across panels and which change meaning. An additional schematic of the model in this figure would help with clarity of the presentation.

We have redesigned Figure 1 and made two major changes:

7. In Equation 1, is P used at all? If not simplify. The explanation of P sounds complicated without really saying anything specific. Give an example of constraints that could be incorporated.

We have removed mention of *P* from the Results section, as it is not needed to understand the main arguments and its choice does not affect allocation. We have kept it in the Methods section, to acknowledge that whitening is not unique and because it can be used to develop more complex models that enforce further constraints, but still allow a straightforward determination of the allocation. We now give a specific example of how such a constraint (in this case sparsity) could be enforced:

“As the optimally whitened solution is independent of P, additional constraints on the solution can be enforced. For example, power constraints are often added, which either limit the total variance or equalize the variance across output neurons, and additional sparsity constraints can be placed on the output neuron’s activity or their receptive field weights (Doi et al., 2012; Doi and Lewicki, 2014). For example, to optimize the sparsity of the weight matrix, one would define an appropriate cost function (such as the L1 norm of the weight matrix), then iteratively calculate the gradient with respect to W , take an update step to arrive at a new W_opt, and determine P as described above in Equation 7 (see Doi and Lewicki, 2014, for further details). Importantly for our problem, such additional constraints will affect the precise receptive field structure (through P), but not the eigenvalues and eigenvectors […].”

8. The assumption that the covariance between regions is zero is quite a strong one and seems central to the paper. This should be justified or at least discussed how realistic it is. In the finger tip example, one of course typically touches objects not with one finger but with multiple ones.

If covariance functions are monotonically decreasing and they are doing so fast enough that the extent of correlations is smaller than the size of the different regions, then the block matrix approximation is very accurate (as we have shown previously numerically). Of course, covariance functions might NOT decrease monotonically and, as pointed out here, one might expect this to be the case in touch where multi-finger contact would induce long-ranging and nonlinear correlations.

We have responded to this point by expanding the corresponding paragraph in the ‘Limitations’ section of the Discussion:

“We considered allocations for monotonically decreasing covariance functions. Choosing a negative exponential function and assuming that activations are uncorrelated across different input regions allowed us to derive an analytical solution. How justified are these assumptions? Many sensory systems will likely obey the intuitive notion that receptor correlations decrease with distance; however, there are notable exceptions, for example in the auditory system (Terashima and Okada, 2012). In touch, the sensory system we are mainly concerned with here, contact might often be made with several fingers (for humans) or rays (for the star-nosed mole) simultaneously, which might induce far-ranging non-monotonic correlations. Unfortunately there is little quantified evidence on the strength of these correlations, rendering their importance unclear. At least for the star-nosed mole, their prey is often small compared to the size of the rays, which move mostly independently, so cross-ray correlations might be low. Furthermore, the receptive fields of neurons in early somatosensory cortex of both humans and star-nosed moles are strongly localized to a single appendage and lie within cortical sub-regions that are clearly delineated from others (Sur et al., 1980; Nelson et al., 1980; Catania and Kaas, 1995). If long-range non-monotonic correlations were strong, we would expect to find many multi-appendage receptive fields and blurry region boundaries. As this is not the case, it therefore stands to reason that either these correlations are not very strong or that there is some other process, perhaps during development, that prevents these correlations affecting the final allocation. Either way, our assumption of monotonically decreasing covariance functions appears to be a good first-order match. That said, the question of how to arrive at robust allocations for more complex covariance functions is an important one that should be considered in future research. While it is possible in principle to solve the allocation problem numerically for arbitrary covariance functions, in practice we noticed that the presence of small numerical errors can affect the sorting process and caution is therefore warranted, especially when considering non-monotonic functions.”

9. The brackets in Figure 3 were found to be very confusing, please revise.

We have changed the brackets to markers whose colors match those in panel B.

10. The covariance functions in Figure 5 are not very different, it is not surprising that these result in very similar outcomes. Comment on this or consider revising.

We agree that the results are not particularly surprising, however they demonstrate the robustness of the method (i.e. the results are not an artefact of assuming a negative exponential covariance function) and provide some additional, insightful context on how the particular covariance structure affects the resulting allocation, tested via a systematic change in the smoothness of the function. Last but not least, the covariance functions tested were specifically requested by reviewers in the previous revision round. We have now clarified the motivation behind including these functions and what we can learn from them at the start of the relevant section:

“The results above relate to negative exponential covariance functions, where allocations can be solved analytically and it is unclear whether these findings generalize to other monotonically decreasing covariance functions and are therefore robust. Specifically, negative exponential covariances are relatively 'rough' and allocations might conceivably depend on the smoothness of the covariance function. To test directly how optimal allocations depend systematically on the smoothness of the covariance function, we numerically calculated allocations for three covariance functions […]”